# Distinct filament morphology and membrane tethering features of the dual FtsZ paralogs in Odinarchaeota

Jayanti Kumari [1,9], Akhilesh Uthaman [1,9], Sucharita Bose [2,10], Ananya Kundu [3,4,10], Vaibhav Sharma [5,10], Soumyajit Dutta [6], Anubhav Dhar[1], Srijita Roy [7,8], Ramanujam Srinivasan [7,8], Samay Pande [5], Kutti R Vinothkumar [3], Pananghat Gayathri [6✉] & Saravanan Palani [1✉]

## Abstract

The Asgard phylum has emerged as a model to study eukaryogenesis because of their close relatedness with the eukaryotes. In this study, we use FtsZ proteins from a member of the class Odinarchaeia as representatives to investigate the probable origin, evolution, and assembly of the FtsZ/tubulin protein superfamily in Asgard archaea. We performed a comparative analysis of the biochemical properties and cytoskeletal assembly of FtsZ1 and FtsZ2, the two FtsZ isoforms in the Odinarchaeota metagenome. Our electron microscopy analysis reveals that OdinFtsZ1 assembles into curved single protofilaments, while OdinFtsZ2 forms stacked spiral ring-like structures. Upon sequence analysis, we identified an N-terminal amphipathic helix in OdinFtsZ1, which mediates direct membrane tethering. In contrast, OdinFtsZ2 is recruited to the membrane by the anchor OdinSepF via OdinFtsZ2's C-terminal tail. Overall, we report the presence of two distant evolutionary paralogs of FtsZ in Odinarchaeota, with distinct filament assemblies and differing modes of membrane targeting. Our findings highlight the diversity of FtsZ proteins in the archaeal phylum Asgardarchaeota, providing valuable insights into the evolution and differentiation of tubulin-family proteins.

**Keywords** Asgardarchaeota; Eukaryogenesis; FtsZ; Membrane; Odinarchaeota
**Subject Categories** Cell Adhesion, Polarity & Cytoskeleton; Evolution & Ecology; Structural Biology

## Introduction

Asgardarchaeota, a newly identified archaeal phylum (hereafter referred to as the Asgard phylum), has emerged as the closest relative of eukaryotes (Spang et al, 2015; Zaremba-Niedzwiedzka et al, 2017; Eme et al, 2017; Spang et al, 2018; MacLeod et al, 2019; Williams et al, 2019; Liu et al, 2021; Eme et al, 2023; Vosseberg et al, 2024; Zhang et al, 2025). Metagenome Assembled Genome (MAGs) of these archaea revealed the presence of multiple Eukaryotic Signature Proteins (ESPs) such as cell division-associated proteins (Spang et al, 2015; Eme et al, 2023), trafficking proteins (Neveu et al, 2020; Vargová et al, 2025), membrane remodeling (Lu et al, 2020; Hatano et al, 2022; Souza et al, 2025; Melnikov et al, 2025), protein translocation (Carilo et al, 2024) and glycosylation-related proteins (Nakagawa et al, 2024). The identification of cytoskeletal protein homologs like Lokiactin, gelsolin, profilin, homologs of subunit 4 of the Arp2/3 complex, tubulin-like proteins in various genomes provides evidence for the archaeal origin of the present-day eukaryotic cytoskeleton (Akıl and Robinson, 2018; Akıl et al, 2020; Stairs and Ettema, 2020; Akıl et al, 2022; Wollweber et al, 2025). An interesting feature of Asgard archaea is the presence of cytoskeletal elements such as FtsZ and SepF, typically associated with prokaryotes, alongside tubulin, ESCRT and actin, commonly associated with eukaryotes. This raises important questions about their organization and role in processes such as cell division.

FtsZ, a tubulin superfamily protein, is an essential player in cell division in many bacteria and archaea (Bi and Lutkenhaus, 1991; Erickson et al, 2010; Margolin, 2005; Liao et al, 2021). In bacteria, overlapping FtsZ filaments assemble as Z-rings at the cell midbody and help recruit other divisome proteins (Li et al, 2007). Tread-milling filaments of FtsZ form the leading edge of the invaginating septum and direct inward growth of the septal wall to ensure the successful separation of cells (Margolis and Wilson, 1981; Li et al,

[1]Department of Biochemistry, Indian Institute of Science, Bengaluru, Karnataka 560012, India. [2]Institute for Stem Cell Science and Regenerative Medicine, Bengaluru, Karnataka, India. [3]National Center for Biological Sciences, Tata Institute of Fundamental Research, GKVK Post, Bengaluru, Karnataka 560065, India. [4]School of Biotechnology, Amrita Vishwa Vidyapeetham, Amritapuri, Kollam, Kerala 690525, India. [5]Bacterial Ecology and Evolution group, Department of Microbiology and Cell Biology, Indian Institute of Science, Bengaluru, Karnataka, India. [6]Biology Division, Indian Institute of Science Education and Research, Pune, Maharashtra, India. [7]National Institute of Science Education and Research, Bhubaneswar, Odisha, India. [8]Homi Bhabha National Institutes (HBNI), Training School Complex, Anushakti Nagar, Mumbai, Maharashtra 400094, India. [9]These authors contributed equally as first authors: Jayanti Kumari, Akhilesh Uthaman. [10]These authors contributed equally: Sucharita Bose, Ananya Kundu, Vaibhav Sharma. ✉E-mail: gayathri@iiserpune.ac.in; spalani@iisc.ac.in

2007; Bisson-Filho et al, 2017; Yang et al, 2017; McCausland et al, 2021; Whitley et al, 2021). FtsZ filament formation is GTP-dependent, and results in a GTP hydrolysis site at the interface of two monomers, (De Boer et al, 1992; Löwe and Amos, 1998; Mukherjee and Lutkenhaus, 1998; Lu et al, 2000; Oliva et al, 2003) as is the case with α/β-tubulin (Erickson, 1995, 1997; Nogales et al, 1998a, 1998b). However, the kinetic polarity of FtsZ filaments is expected to be opposite to that of tubulin, with the growing end being the bottom end (defined as the exposed interface consisting of C-terminal Domain (CTD) away from the nucleotide-binding pocket) (Du et al, 2018; Ruiz et al, 2022; Chakraborty et al, 2024). Several studies including electron cryotomography of *Caulobacter crescentus* cells revealed arc-like FtsZ filaments at mid-cell (Li et al, 2007). Additionally, investigation of in vitro reconstituted FtsZ filaments inside liposomes showed the presence of ring-like structures, which cause membrane deformation upon GTP hydrolysis (Osawa et al, 2008; Osawa and Erickson, 2013; Szwedziak et al, 2014). This ring is anchored to the membrane via another divisome protein called FtsA, an actin homolog (Bork et al, 1992; Pichoff and Lutkenhaus, 2005; Szwedziak et al, 2012). In addition to FtsA, many Gram-negative bacteria including *Escherichia coli* possess additional FtsZ membrane anchors like ZipA (Hale and de Boer, 1997; Pichoff, 2002), whereas Gram-positive bacteria and cyanobacteria possess a different anchor called SepF. SepF has been reported to bind FtsZ and interestingly is conserved in archaeal systems as well (Hamoen et al, 2006; Duman et al, 2013; Pende et al, 2021; Nußbaum et al, 2021).

Unlike most bacteria and archaeal superphyla such as Euryarchaeota and DPANN, which depend solely on FtsZ for cell division (Margolin, 2005; Erickson et al, 2010; Liao et al, 2021), the TACK superphylum predominantly relies on ESCRT homologs (Nußbaum et al, 2024; Lindås et al, 2008; Samson et al, 2008; Hurtig et al, 2023). Archaeal FtsZ-based cell division, outside of Asgard phylum, depends on either one or two forms of FtsZ (Liao et al, 2021; Santana-Molina et al, 2023), with SepF acting as a membrane anchor (Pende et al, 2021; Ithurbide et al, 2022). The archaeal cell division mechanism and the key factors involved remain largely unexplored.

An Asgard archaeon, *Candidatus* Odinarchaeum yellowstonii LCB_4 (Class Odinarchaeia) (hereafter referred to as Odinarchaeota) uniquely possesses both, a tubulin-like gene and two FtsZ genes (Zaremba-Niedzwiedzka et al, 2017; Tamarit et al, 2022; Akıl et al, 2022; Ithurbide et al, 2022). Recent structural study on OdinTubulin and comparison to eukaryotic α/β-tubulins has suggested the possible evolution of tubule-like structures via stabilization of inter-protofilament interactions (Akıl et al, 2022). Another study on *Candidatus* Lokiarchaeum ossiferum (Class Lokiarchaeia) suggests a pre-eukaryotic origin of microtubules in the Asgard archaeon (Wollweber et al, 2025). The presence of a tubulin-like gene has also been reported in another environmental metagenome of a Heimdallarchaeon (preprint: Baker et al, 2023; Wollweber et al, 2025). This intriguing co-existence of FtsZ and tubule-forming tubulin, among Asgard archaea raises interesting questions about the origins of eukaryotic tubulin, their adaptation into chromosome segregation, cytoskeletal arrangement and their respective contributions to different cellular processes. The study of Asgard cytoskeletal proteins, thus, holds the potential for unveiling new insights into structural diversity, functional adaptations, crosstalk and coordination among cytoskeletal protein families

during the course of evolution. Although the structure of OdinTubulin is known, the structural features and functional specializations of the two FtsZ proteins in Odinarchaeota remain completely unknown. Moreover, the unique presence of dual FtsZ genes in Odinarchaeota among other members of the Asgard phylum raises questions about their function and similarity to other archaea that utilize dual FtsZ genes for cell division. Diversification of function has also occurred during evolution among other members of FtsZ/tubulin family of proteins. For example, TubZ (Larsen et al, 2007) and PhuZ (Chaikeeratisak et al, 2017; Erb et al, 2014) are involved in DNA segregation and maintenance in bacteria and bacteriophages, respectively. On the contrary, the CetZ family of proteins that are commonly found in many archaea, play a role in cell shape maintenance and do not affect cell division processes (Duggin et al, 2015). Further, MinD, a spatial regulator of Z-ring assembly in *E. coli*, has been co-opted in archaea to determine the position of the motility machinery (Nußbaum et al, 2020). This led us to carry out studies on the two candidate FtsZ genes in Odinarchaeota to characterize their biochemical properties and filament structures, and to gain insights about their function.

Here, we present evidence that two FtsZ-like genes in Odinarchaeota assemble into distinct filament morphologies in the presence of GTP. OdinFtsZ1 assembles into the typical FtsZ single protofilament structure, whereas OdinFtsZ2 possesses a curved spiral ring-like filament morphology. We show that SepF, a characteristic membrane anchor for FtsZs, specifically binds OdinFtsZ2 via the C-terminal variable region, whereas OdinFtsZ1 associates with membranes via its hydrophobic N-terminal region that enables direct membrane binding independent of OdinSepF. Overall, our study provides insights into the biochemical and structural features of the dual FtsZs in Asgard archaea, which will aid in understanding their potential roles in the cell division apparatus. Our observations offer implications for the origin and diversification of the FtsZ/tubulin superfamily across the different domains of life.

## Results

### *Candidatus* Odinarchaeum encodes two distinct FtsZ paralogs

A phylogenetic analysis of representative FtsZ protein sequences from the bacterial and archaeal domains identified two candidate sister clades corresponding to two distinct clusters of FtsZ proteins confirming the ancestral duplication and diversification of FtsZ in archaea, prior to the emergence of Asgard lineage. The two candidate FtsZ proteins from Odinarchaeota clustered along with the other FtsZ1 and FtsZ2 proteins present in archaeal domain (Fig. EV1). AlphaFold (Jumper et al, 2021) prediction of monomeric structures of Odin FtsZs (UniProt Accession: OdinFtsZ1 (A0A1Q9N645), OdinFtsZ2 (A0A1Q9N6K6)) shows that both the candidate proteins, like bacterial FtsZ (*Escherichia coli* FtsZ; EcFtsZ; PDB ID: 6UMK) and the archaeal FtsZ (*Methanococcus jannaschii* FtsZ; MjFtsZ1; PDB ID: 1FSZ), possess an N-terminal domain (NTD), which contains the nucleotide-binding pocket and a C-terminal domain (CTD) with the H7 helix positioned between the two domains (Fig. 1A). The end of this helix carries a loop (T7 loop) containing the catalytic residues important for GTP hydrolysis (Fig. 1A). The presence of highly

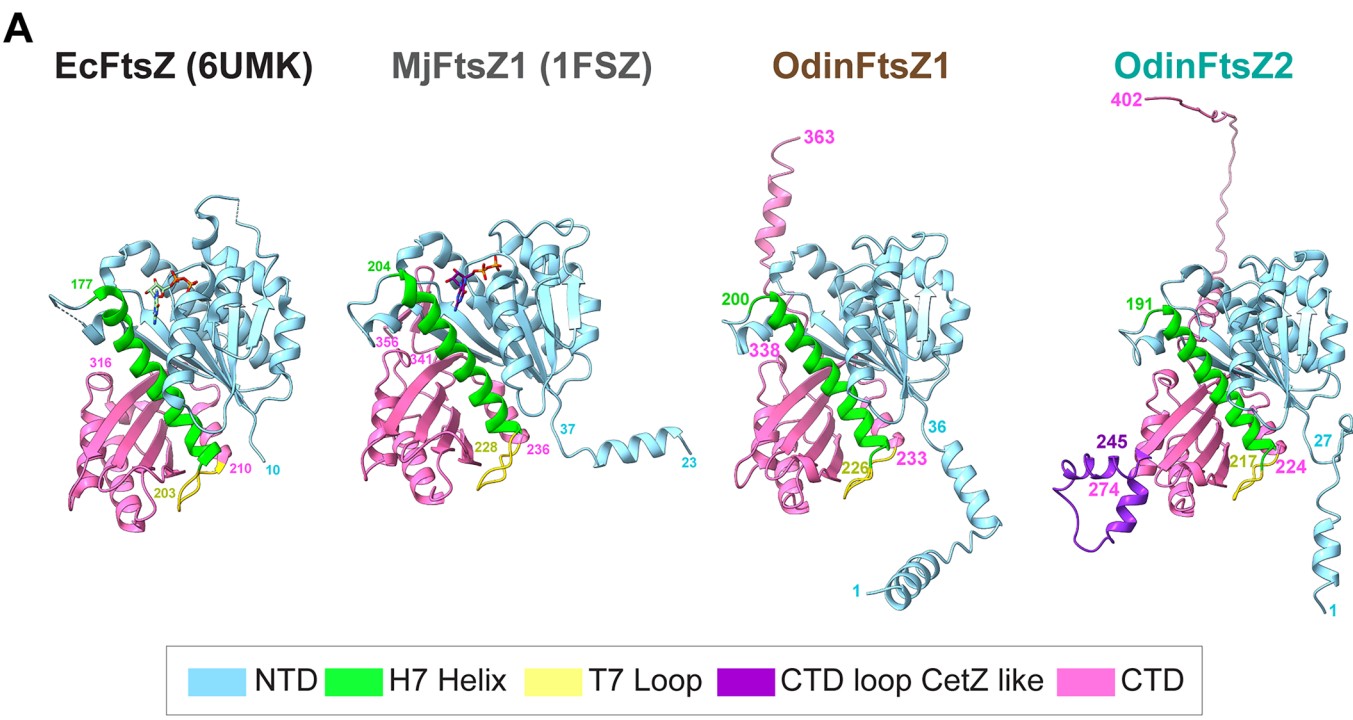

**A**

EcFtsZ (6UMK)  MjFtsZ1 (1FSZ)  OdinFtsZ1  OdinFtsZ2

Legend:
- NTD
- H7 Helix
- T7 Loop
- CTD loop CetZ like
- CTD

**B**

GGGTGTG

GGGTG(T/S)G

NxDxx(D/E)

| | | | | | | |
|---|---|---|---|---|---|---|
| Mj.FtsZ1 ... | 129 | CGL GGGTGTC SAP | 141 | ... 230 | GLI NVD FAD VKAVMNNGG | 247 ... |
| Odin FtsZ1 ... | 126 | CGL GGGTGTC AAP | 138 | ... 227 | GLI NLD FAD VKKVMNNKG | 244 ... |
| Odin FtsZ2 ... | 118 | AGM GGGTGTC SAP | 130 | ... 218 | SLI NLD YAD VRTIMTNGG | 235 ... |
| Ec.FtsZ ... | 102 | AGM GGGTGTC AAP | 131 | ... 204 | GLM NVD FAD VRTVMSEMG | 221 ... |

T3    T7    H8

**C**

OdinFtsZ1(5μM)
OdinFtsZ1(5μM) + 5mM Mg$^{2+}$
OdinFtsZ1(5μM) + 5mM Mg$^{2+}$ + 2mM GTP

Relative Intensity (A.U.) vs Time (sec)

**D**

OdinFtsZ2(5μM)
OdinFtsZ2(5μM) + 5mM Mg$^{2+}$
OdinFtsZ2(5μM) + 5mM Mg$^{2+}$ + 2mM GTP

Relative Intensity (A.U.) vs Time (sec)

◀ **Figure 1.** *Candidatus* Odinarchaeum yellowstonii LCB_4 carries two paralogs of polymeric FtsZ proteins.

(A) Structural comparison of protomer structures of *Escherichia coli* FtsZ (EcFtsZ; PDB ID: 6UMK), *Methanococcus jannascchi* FtsZ1 (MjFtsZ1; PDB ID:1FSZ) with the predicted AlphaFold monomeric structures for OdinFtsZ proteins, OdinFtsZ1: and OdinFtsZ2). The N-terminal domain (NTD) of the structure is colored in cyan, the H7 helix in green, the T7 loop in yellow, the C-terminal domain (CTD) of the FtsZ monomer in pink, and the extra loop insertion of OdinFtsZ2 in purple. (B) Representative sequence logo for the tubulin signature motifs: GTP binding (left) and GTP hydrolysis motif (right). The WebLogo is prepared using a sequence alignment of 69 FtsZ sequences used in the preparation of the phylogenetic tree described in (Fig. EV1). The alignment shown in the figure has four representative sequence whose monomer structure is represented in (A) namely *Methanococcus jannaschii* FtsZ1 (UniProt Accession: Q57816, *Escherichia coli* FtsZ (UniProt Accession: P0A9A6), OdinFtsZ1(UniProt Accession: A0A1Q9N645) and OdinFtsZ2 UniProt Accession: A0A1Q9N6K6). (C) The light scattering profiles plotted as relative light scattering intensity (y axis) with time (x axis) for OdinFtsZ1 protein (5 μM) polymerized in the presence of 2 mM GTP and 5 mM $Mg^{2+}$ (red), only 5 mM $Mg^{2+}$ without GTP (green), and without the addition of $Mg^{2+}$ or GTP (purple). (D) The corresponding plots are as in (C) for the OdinFtsZ2 protein (5 μM). The shaded region around the individual plots in (C, D) denote the standard deviation (SD) over three independent repeats. Source data are available online for this figure.

conserved GTP binding (GGGTG[T/S]G) and GTP hydrolysis (NxDxx[D/E]) motifs (Vaughan et al, 2004) in these OdinFtsZ sequences indicates conservation of crucial functional elements related to GTP binding and hydrolysis across both bacterial and archaeal domains (Fig. 1B). Though OdinFtsZ2 clusters along with other FtsZ2 members (Fig. EV1), it possesses an insertion of a loop in the CTD, which is not found in other FtsZ proteins, but is surprisingly a characteristic feature of the archaeal CetZ family proteins (Fig. 1A; Appendix Fig. S1A,B).

FtsZ, a GTPase family protein, polymerizes in a GTP-dependent manner (RayChaudhuri and Park, 1992; Erickson et al, 1996). To monitor the polymerization of purified OdinFtsZ1 and OdinFtsZ2 proteins (Appendix Fig. S2A), we employed 90 °C angle light scattering (Mukherjee and Lutkenhaus, 1999). For the OdinFtsZ proteins, there was an increase in scattering upon the addition of GTP. Interestingly, the scattering profiles indicate differences in assembly dynamics between the two Odin FtsZs. Notably, there was an increase in scatter observed in OdinFtsZ1 in the presence of $Mg^{2+}$ ion (5 mM), even without any supplemented nucleotide (Fig. 1C), a property not observed for OdinFtsZ2 (Fig. 1D). The maximum relative intensity of polymerized OdinFtsZ1 was achieved faster than OdinFtsZ2, followed by a gradual decrease (Fig. 1C,D; Appendix Fig. S2B,C). We speculate that the observed variation in light scattering associated with FtsZ polymerization might be linked to polymer mass (Mukherjee and Lutkenhaus, 1999), distinct filament morphologies, or assembly dynamics for OdinFtsZ1 and OdinFtsZ2 proteins.

## OdinFtsZ1 forms a single protofilament assembly

The difference in the scattering pattern between the two proteins suggests potential disparities in filament assembly. Therefore, we assessed the ability of these proteins to form filaments in the presence of nucleotides. To further understand how different nucleotide states affect the polymerization of these filaments, we tested the assembly of OdinFtsZ filaments in the presence of different nucleotide analogs using negative staining and transmission electron microscopy (TEM). We observed filament formation for OdinFtsZ1 in the presence of GTP, non-hydrolysable GTPγS, GMPCPP (a slow-hydrolyzing nucleotide analog) and GDP (Appendix Fig. S2D). Filaments were observed for OdinFtsZ1 even without added nucleotide (Appendix Fig. S2D). For cryo-EM analysis, we incubated OdinFtsZ1 with 2 mM GMPCPP on ice to promote the formation of relatively straight filaments. This enabled visualization of thin filaments (Fig. 2A), resembling the single protofilaments typically seen in other canonical FtsZs (Romberg

et al, 2001; González et al, 2005; Wagstaff et al, 2017; Fujita et al, 2023). The filaments were extracted from the micrographs and 2D class averaged (Fig. 2B). After refining the helical parameters (Fig. EV2A), the 3D map was constructed that represents a single protofilament of OdinFtsZ1 (Fig. 2C) with a 44.5 Å helical rise and a −2.72° helical twist (Fig. EV2A). Although the overall resolution of the final reconstructed 3D map is ~3.6 Å (FSC = 0.143) (Fig. EV2A), the map quality was poor for reliable main-chain tracing in some regions, and the resolution estimated from the atomic model (dmodel or map vs model at FSC 0.5) was limited to 4.1 Å (Fig. EV2B,C). The map revealed the canonical FtsZ/tubulin fold (Löwe and Amos, 1998; Löwe, 1998), featuring an N-terminal nucleotide-binding domain (NTD) and a C-terminal GTPase activating domain (or CTD). The CTD is composed of a four-stranded β-sheet flanked by two α-helices on each side with a long central helix (H7) connecting the two domains (Fig. EV3A), consistent with the AlphaFold structure prediction (Fig. 1A). The N-terminal domain consists of six β-strands arranged into a sheet, flanked by two α-helices on one side and three on the other. The map resolution enabled clear visualization of the individual β-strands, the α-helices, and the central connecting helix (Figs. 2D and EV3A).

FtsZ subunits are known to adopt two distinct conformations: an open state that exhibits high filament affinity (Tense or T-state) and a closed state characterized by low filament affinity (Relaxed or R-state) (Fujita et al, 2017). This conformational switch is driven by polymerization and depolymerization rather than the nucleotide-binding state of the monomers, which is essential for protofilament cooperative assembly and treadmilling (Wagstaff et al, 2017). The cryo-EM structure of OdinFtsZ1 (PDB ID: 9V7V) reveals a protofilament with the monomer subunits in the T-conformation similar to the cryo-EM structures of *Klebsiella pneumoniae* FtsZ (KpFtsZ) (Fujita et al, 2023) and EcFtsZ (Wagstaff et al, 2017). The OdinFtsZ1 protofilament adopted the T-conformation, featuring an extended interface area of 1183 $Å^2$ as calculated using PISA web server (Krissinel and Henrick, 2007)—comparable to that previously observed for *Staphylococcus aureus* FtsZ (SaFtsZ; PDB ID: 3VOA) and KpFtsZ (PDB ID: 8IBN). The primary difference between the two conformations lies in the H7 helix, which shifts downward by one helical turn in the T-conformation, pushing the T7 loop into the subunit interface where it interacts with the nucleotide bound to the adjacent (lower) subunit (Fig. EV3B). This conformational change is accompanied by a 25–27° rotation of the C-terminal domain (Matsui et al, 2012). Structural superposition of OdinFtsZ1 with SaFtsZ, which adopts the R conformation (PDB ID: 5H5G; chain B), underscores these differences (Fig. EV3B).

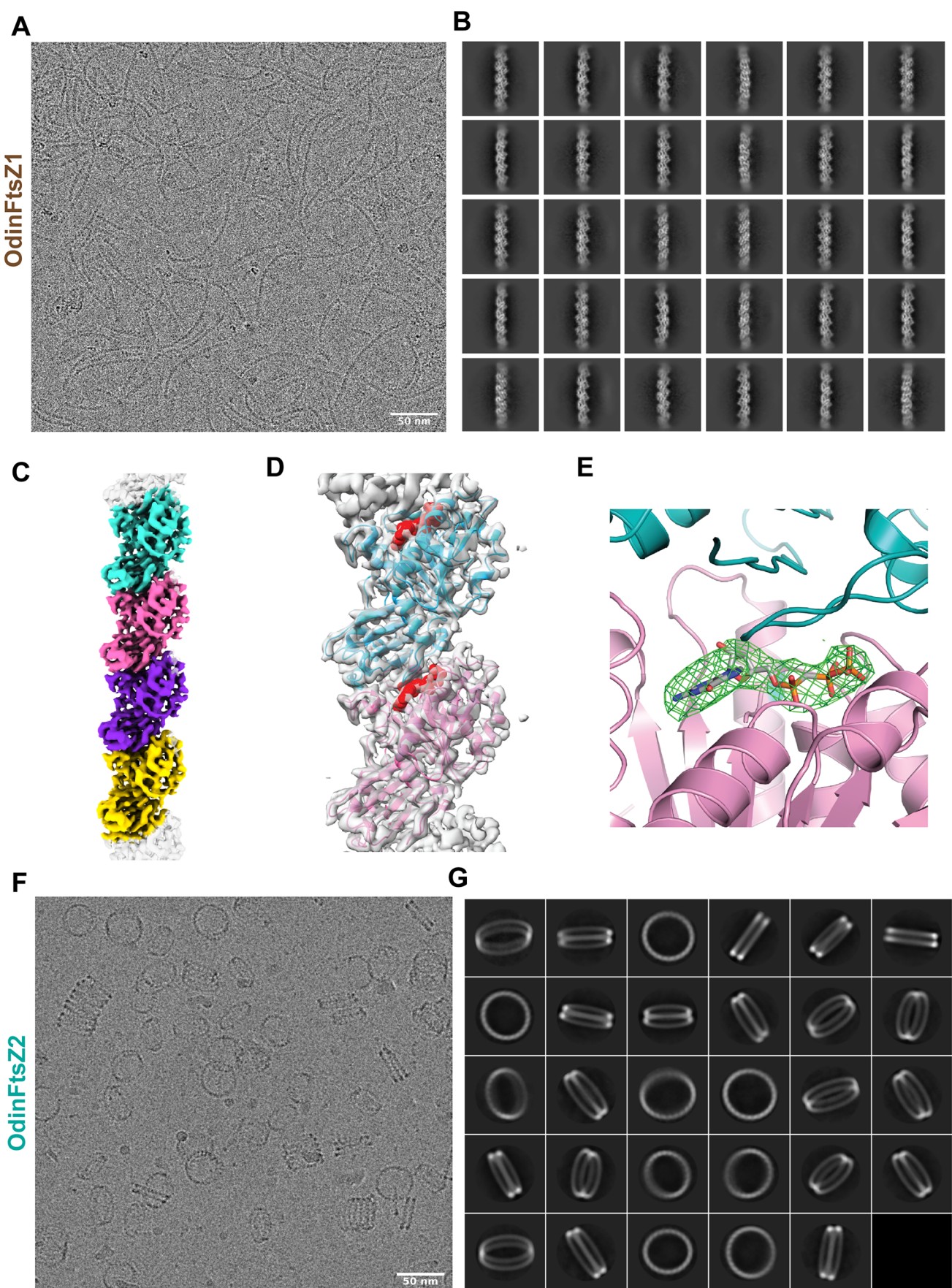

Figure 2. **OdinFtsZ proteins assemble into distinct filament morphologies.**

(A) cryo-EM micrograph of OdinFtsZ1 (0.4 mg/mL) polymerized in the presence of 2 mM GMPCPP (scale bar—50 nm). (B) Representative 2D class averages of OdinFtsZ1 showing high-resolution features. The box size is 256 pixels, corresponding to 273.9 Å. (C) Final sharpened map of OdinFtsZ1 contoured at threshold level 0.424 in ChimeraX. Each monomer is shown in a different color based on the boundary derived from the model that was built. The top and bottom edges of the map are not modeled and shown in gray. (D) OdinFtsZ1 model fitted into the final sharpened map. The sharpened map is shown in gray, while the fitted atomic model is displayed in cartoon representation with the two monomers colored in pink and cyan. GMPCPP is colored in red and represented as a space-filled model. (E) Identification of the non-protein density, i.e., bound GMPCPP molecule in OdinFtsZ1 using the difference ($F_o$-$F_c$) density map, shown in green, as calculated by Servalcat (Yamashita et al, 2021). (F) cryo-EM micrograph of OdinFtsZ2 (4 mg/mL) polymerized with 2 mM GTP (scale bar—50 nm). (G) Representative 2D class averages of OdinFtsZ2 from particles that forms the shorter stacks of two rings. The box size is 540 pixels, corresponding to 577.8 Å. The diameter of the double ring is ~360 Å. Source data are available online for this figure.

A non-protein density was detected within the nucleotide-binding pocket using Servalcat (Yamashita et al, 2021) into which GMPCPP was modeled (Fig. 2D,E). The phosphate binding loops T1 (residues 39–45), T2 (residues 65–68), T3 (residues 90–96) and T4 (residues 124–130) (Löwe J, 1998) surround the nucleotide GMPCPP. At this resolution, the side chains of most residues are not well resolved in the density map, limiting our ability to make definitive conclusions about specific molecular interactions. Although $Mg^{2+}$ was included in the incubation mixture, no corresponding density was observed. Additional density near the N-terminus of the monomer was observed but could not be reliably modeled due to the limited map resolution (Fig. EV3C). Super-position of the AlphaFold-predicted structure (UniProt Accession: A0AAF0ID74) onto the map suggests this to be a portion of the N-terminal segment of the protein (residues 1–33).

## OdinFtsZ2 assembles into spiral ring-like structures and co-pellets with OdinFtsZ1

Interestingly, OdinFtsZ2 formed highly curved, stacked ring-like structures when provided with GTP or GTPγS while showing very few higher-order filament assemblies in the presence of GMPCPP and GDP and no nucleotide condition (Appendix Fig. S2E). This observation suggests a role for nucleotide binding for the assembly of OdinFtsZ2 and consistent with the absence of an increase in light scattering when there was no nucleotide added (Fig. 1D). cryo-EM of OdinFtsZ2 protein in complex with GTP revealed that the filaments self-assembled into structures with high curvature, resembling stacked spring-like assemblies (Fig. 2F). A key observation is the formation of a variable number of rings in the stacks by OdinFtsZ2 in the current experimental conditions (Fig. 2F), similar to OdinTubulin (Akıl et al, 2022). In each field of view, the number of ring-like structures in each assembly varied. Assemblies with multiple stacked rings were also seen often, but those with two stacked rings were more prominent. These were chosen and averaged in 2D (Fig. 2G). The low resolution of the ring shows no clear indication of subunit stoichiometry within a pair of rings (Fig. 2F,G). Factors promoting the stable formation of stacked rings are currently being investigated.

To assess if OdinFtsZ proteins interacted with each other, we utilized polymer pelleting or sedimentation assays. FtsZ polymer-ization and subsequent pelleting were first tested individually for OdinFtsZ1 and OdinFtsZ2, both in the presence and absence of GTP. Again, consistent with the light scattering and TEM micrographs, we observed a substantial pellet fraction for OdinFtsZ1 without supplementing with any nucleotide (Fig. EV4A). However, GTP addition significantly enhanced the amount of OdinFtsZ1 in the pellet fraction, suggestive of a filament assembly stimulated in the presence of nucleotide (Fig. EV4B). However, OdinFtsZ2 was not observed in the pellet fraction, irrespective of the presence or absence of GTP (Fig. EV4A,B). It is interesting to note that OdinFtsZ2 does not sediment despite the formation of higher-order assemblies confirmed by TEM and scattering. This might be due to the higher frictional ratio of sedimentation for the stacked ring-like assemblies or the absence of filament bundling in a spring-like assembly. FtsZ filaments are known to pellet more efficiently in a sedimentation assay when they have a higher propensity to form bundles (Król and Scheffers, 2013).

Since OdinFtsZ2 failed to pellet when incubated with GTP, we could use this assay to test the probable co-pelleting of OdinFtsZ2 in the presence of OdinFtsZ1. When both the Odin FtsZs were incubated at equimolar concentrations in the presence of GTP, a significant fraction of OdinFtsZ2 was seen in the pellet fraction along with OdinFtsZ1 (Fig. EV4A–C). This co-sedimentation was dependent on the presence of nucleotides, suggestive of an interaction, probably between the filament states of the two FtsZ homologs. This interaction between OdinFtsZ1 and OdinFtsZ2 observed in vitro suggests a likely interaction in vivo as well as coordinated cellular function for the FtsZs in Odinarchaeota.

## SepF mediated membrane tethering of OdinFtsZ2

In bacteria, FtsZ proteins are typically tethered to the membranes via interactions through their conserved CCTP motifs with adaptor proteins like FtsA, ZipA and SepF (Hale and de Boer, 1997; Cendrowicz et al, 2012; Pende et al, 2021). While ZipA is an integral membrane protein (Hale and de Boer, 1997), the anchor proteins SepF and FtsA bind the membrane via amphipathic helices at their N-termini and C-termini, respectively (Pende et al, 2021; Cendrowicz et al, 2012). In contrast to FtsA, SepF is widely conserved across archaeal domain (Fig. EV5). It is also known that these proteins localize to the Z-ring and that their depletion causes severe cell division defects (Hamoen et al, 2006; Nußbaum et al, 2021). Recent findings in *Bacillus subtilis* shows that in a minimal divisome, FtsZ and SepF alone suffice to form an active Z-ring (Gulsoy et al, 2025). The consistent presence of SepF in archaeal systems with FtsZ suggests a possible functional link between the two proteins. SepF dimers from bacterial systems like *B. subtilis* are known to oligomerize further (Duman et al, 2013; Choudhury and Chaudhuri, 2025). Residues important for these interactions are not conserved in other bacterial species such as *Corynebacterium glutamicum* (Sogues et al, 2020) and archaeal SepF (Pende et al, 2021). In vitro studies support no further oligomeric assembly for archaeal SepF, and thus their oligomerization status remains to be tested.

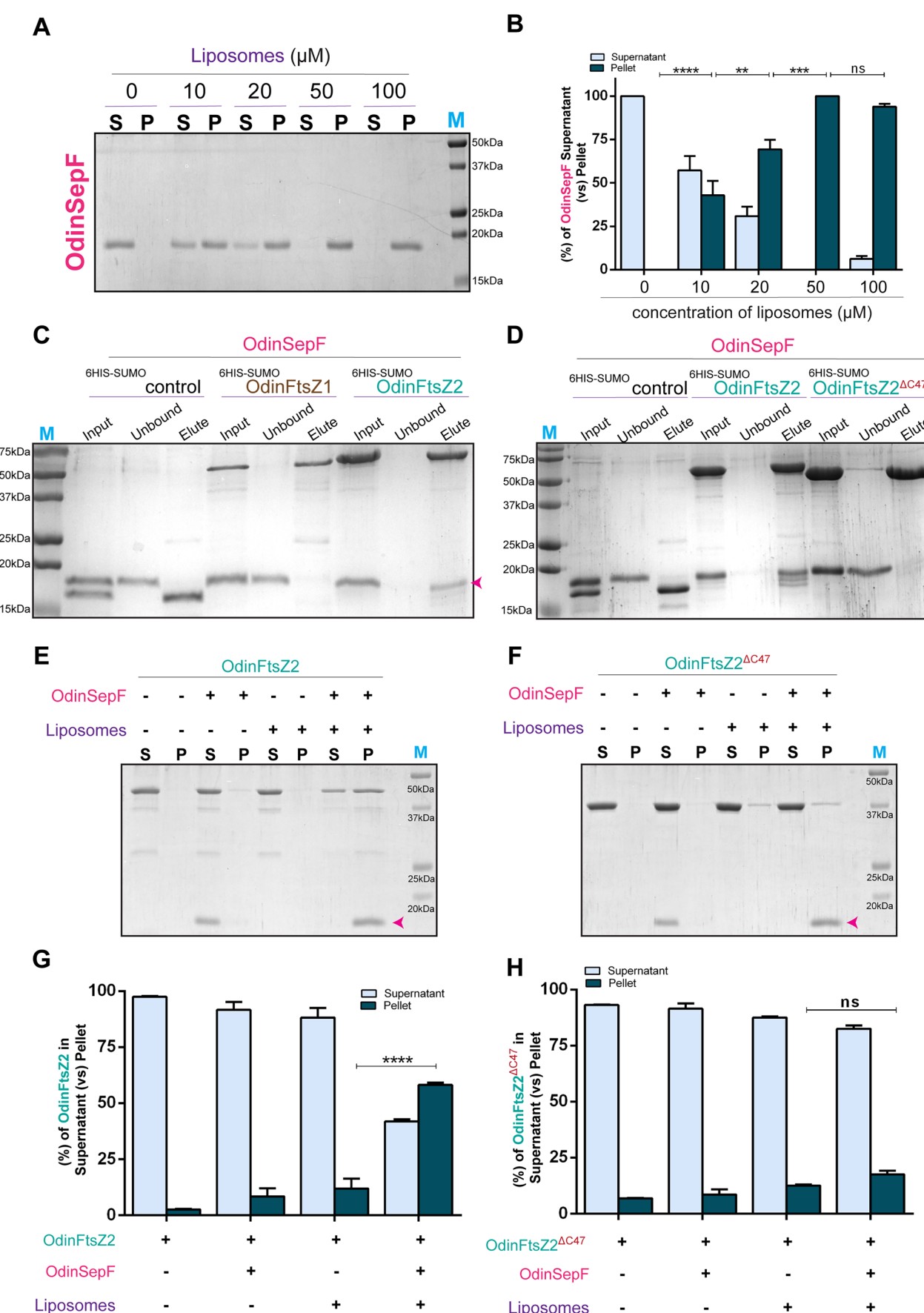

**Figure 3. OdinFtsZ2 is recruited to the liposome via OdinSepF, whereas OdinFtsZ1 shows no interaction with OdinSepF.**

(A) Representative 12% SDS-PAGE gel for OdinSepF sedimentation with increasing liposome concentrations. The supernatant (S) and pellet (P) fractions were separated and loaded upon sedimentation at 100,000×g. Lanes 1 and 2 represent S and P for only OdinSepF without liposomes. Liposomes were made of dioleoyl phosphatidylglycerol (DOPG) at the concentrations labeled above the lanes. (B) Representative plot for the percentage fraction of OdinSepF in the supernatant (S) and pellet (P) fractions with increasing concentrations of liposomes as represented in (A). The intensities of the OdinSepF band in the supernatant and pellet fractions were calculated as a percentage: the intensity of a given fraction divided by the sum of the two intensities (supernatant and pellet fractions). These percentages are represented as mean ± standard error of the mean (SEM) in the graph. ($N = 3$ (results replicated over three independent experiment), two-way ANOVA with Tukey's multiple comparisons test was used, ns non-significant, $**P = 0.0044764$, $***P = 0.0009318$, $****P = 0.0000151$, ns non-significant. (C) Representative 15% SDS-PAGE gel of in vitro binding for 6HIS-SUMO, OdinFtsZ1 and OdinFtsZ2. Bead-bound 6HIS-SUMO, OdinFtsZ1 and OdinFtsZ2 were incubated with untagged OdinSepF. Input, wash and elution fractions were loaded after the incubation. The pink arrow denotes the presence of OdinSepF in the elution fraction of OdinFtsZ2. (D) Representative 15% SDS-PAGE gel of in vitro binding for 6HIS-SUMO, OdinFtsZ2 and OdinFtsZ2$^{\Delta C47}$. (E) Representative 12% SDS-PAGE gel for liposome co-sedimentation of OdinFtsZ2 in the presence of OdinSepF. The supernatant (S) and pellet (P) fractions upon sedimentation at 100,000×g were analyzed for OdinFtsZ2 (alone), OdinFtsZ2 + OdinSepF, OdinFtsZ2 + Liposomes, and OdinFtsZ2 + OdinSepF+ Liposomes. The arrow represents the presence of OdinSepF in the respective lanes. (F) Representative 12% SDS-PAGE gel for liposome co-sedimentation of OdinFtsZ2$^{\Delta C47}$ in the presence of OdinSepF. The supernatant (S) and pellet (P) fractions upon sedimentation at 100,000×g were analyzed for OdinFtsZ2$^{\Delta C47}$ (alone), OdinFtsZ2$^{\Delta C47}$ + OdinSepF, OdinFtsZ2$^{\Delta C47}$ + Liposomes, OdinFtsZ2$^{\Delta C47}$ + OdinSepF + Liposomes. (G) Plot representing the percentage fraction (mean ± standard error of the mean (SEM) of OdinFtsZ2 (y axis) in the supernatant (S) and pellet (P) fractions for the experiment described in (E). ($N = 3$ (results replicated over three independent experiment), Two-Way ANOVA with Tukey's multiple comparisons test was used, $****P = 0.00000003$). (H) Plot representing the percentage fraction (mean ± standard error of the mean (SEM)) of OdinFtsZ2$^{\Delta C47}$ (y axis) in the supernatant (S) and pellet (P) fraction for the experiment described in (F). ($N = 3$ (results replicated over three independent experiment), Two-Way ANOVA with Tukey's multiple comparisons test was used, ns non-significant). Source data are available online for this figure.

To validate the conserved role of SepF in Asgard, we investigated the membrane-binding ability of OdinSepF (UniProt Accession: A0AAF0D343). The helical wheel diagram represents a canonical N-terminal amphipathic helix in the OdinSepF protein (Appendix Fig. S3A). We assessed its membrane interaction via liposome-based pelleting. OdinSepF exhibited strong binding to liposomes as observed by a considerable amount in the pellet fractions with increasing liposome concentration during ultracentrifugation (Fig. 3A,B). The oligomeric status of OdinSepF was tested by size-exclusion chromatography with multi-angle light scattering (SEC-MALS). This enabled the accurate mass measurement of OdinSepF corresponding to the molecular weight of the dimeric protein (Appendix Fig. S3B), as observed for other archaeal SepF proteins (Pende et al, 2021).

To check for the interaction between OdinSepF and OdinFtsZ1/Z2, we purified these proteins (Appendix Fig. S3C) and employed a pull-down assay. Our result indicated interaction between OdinFtsZ2 and OdinSepF, while no detectable interaction was observed with OdinFtsZ1 (Fig. 3C). In most archaeal species, this binding is mediated by a nearly conserved GID motif present at the C-terminus of FtsZ (Pende et al, 2021). Unlike OdinFtsZ1, OdinFtsZ2 contains an extra C-terminal stretch, which might possess a motif analogous to the GID motif (Appendix Fig. S3D). To further confirm the role of the C-terminus, we constructed a truncated OdinFtsZ2 lacking the C-terminal region (OdinFtsZ2$^{\Delta C47}$). This construct completely abolished OdinSepF binding, suggesting a specific interaction between OdinSepF and OdinFtsZ2 through the FtsZ2 C-terminus (Fig. 3D). To validate the role of this interaction for OdinFtsZ2 membrane anchoring, we carried out co-sedimentation assays of SepF-OdinFtsZ2 mixtures in the presence of liposomes. OdinFtsZ2, which exhibits no independent binding with liposomes, co-pelleted with liposomes when incubated with its membrane anchor, OdinSepF (Fig. 3E,G). This suggests that OdinFtsZ2 can be indirectly tethered to membranes through its interaction and association with OdinSepF. The deletion of the C-terminal, OdinFtsZ2$^{\Delta C47}$, abolished its interaction with OdinSepF and the protein remained in the supernatant regardless of the presence of liposomes (Fig. 3F,H).

## Direct membrane anchoring via an N-terminal amphipathic helix in OdinFtsZ1

Reports on *Haloferax volcanii* revealed that FtsZ1 is one of the earliest proteins to localize at the future division site. It is hypothesized that FtsZ1 recruits other divisome proteins to the site of cell division, which in turn organizes SepF (Nußbaum et al, 2021). Pull-down assays in these organisms as well as in Odin, support no physical interaction of SepF with FtsZ1. The mechanism by which FtsZ1 assembles at the future division site remains unidentified. Recent studies on Mycoplasma FtsZs have reported the presence of putative membrane targeting sequences (MTS), which are capable of binding liposomes (Dutta et al, 2025). Also, chloroplast FtsZs are known to interact directly with the membrane (An et al, 2024; Liu et al, 2022). To investigate the possibility of a similar mechanism, we analyzed the sequence of OdinFtsZ proteins. This revealed the presence of an N-terminal extension, slightly longer in OdinFtsZ1 compared to that of OdinFtsZ2 (Fig. 1A). The representative helical wheel diagram depicts a potential amphipathic helix at the N-terminal of OdinFtsZ1 (Fig. 4A). We hypothesized that this helix might possess membrane-binding ability. This membrane interaction was tested with liposome pelleting assay where OdinFtsZ1 showed a significant amount in the pellet fraction in the presence of liposomes (Fig. 4B,C). An N-terminal deletion construct, OdinFtsZ1$^{\Delta NTAH}$ lacking the first 13 amino acids abolished the membrane binding, suggesting the presence of an amphipathic helix at the N-terminus which helps in mediating direct membrane attachment (Fig. 4D).

## Discussion

FtsZ is a key protein essential for cell division in bacterial (Bi and Lutkenhaus, 1991; Erickson et al, 2010) and many archaeal systems (Liao et al, 2021; Pende et al, 2021). Over evolutionary time, the role of cell division in eukaryotes has been taken over by actomyosin machinery and ESCRT-like proteins. Additionally,

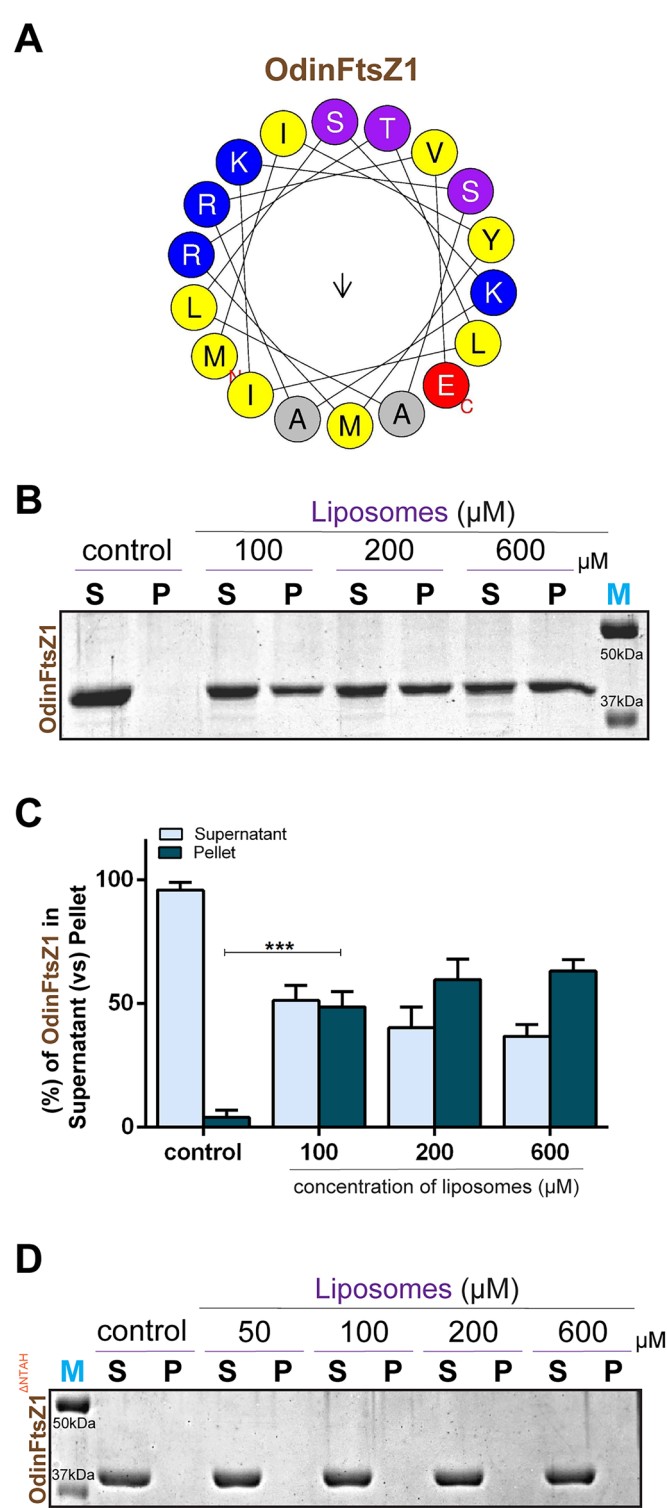

**Figure 4. OdinFtsZ1 binds liposomes via its N-terminal extension.**

(A) Representative helical wheel diagram for the predicted amphipathic helix (AH) at the N-terminus of OdinFtsZ1 screened using Heliquest (version 1.2 Analysis module). (B) Representative 12% SDS-PAGE gel for the liposome-sedimentation of OdinFtsZ1. The supernatant (S) and pellet (P) fractions were separated and loaded upon sedimentation at 100,000×*g*. Lanes 1 and 2 represent S and P for only OdinFtsZ1 without liposomes. (C) Representative plot for the percentage fraction (mean ± standard error of the mean (SEM)) of OdinFtsZ1(*y* axis) in the supernatant (S) and pellet (P) fractions with increasing concentration of liposomes as represented in (B). ($N = 3$ (results replicated over three independent experiment), Two-way ANOVA with Tukey's multiple comparisons test was used, ns non-significant, ***$P = 0.0003$). (D) Representative 12% SDS-PAGE gel for the sedimentation of OdinFtsZ1^ΔNTAH. The supernatant (S) and pellet (P) fractions were separated and loaded upon sedimentation at 100,000×*g*. Lanes 1 and 2 represent S and P for only OdinFtsZ1^ΔNTAH without liposomes. Source data are available online for this figure.

tubulin—a distant cousin of the prokaryotic FtsZ—has diverged to carry out specialized functions such as chromosome segregation (Erickson, 2007; Wickstead and Gull, 2011). Further, organelle-localized FtsZs in eukaryotes function within a complex cytoskeletal environment in association with actin and dynamin-like proteins (Osteryoung and Vierling, 1995; Beech et al, 2000; Gilson et al, 2003; Kiefel, 2004; Miyagishima et al, 2008). Interestingly, some archaeal lineages retain both FtsZ and archaeal homologs of tubulin, highlighting functional divergence among these proteins (Wollweber et al, 2025). Therefore, characterizing the dual FtsZs in Odinarchaeota contributes to a broader understanding of the diversity of FtsZ forms and their functional roles in cytoskeletal organization across domains of life.

Our study aimed to investigate the structural and biochemical features of the dual FtsZs from Odinarchaeota. Our phylogenetic analysis shows that OdinFtsZ1 and OdinFtsZ2 fall into the previously recognized FtsZ1 and FtsZ2 clades, respectively, as seen across multiple archaeal lineages. The placement of these Odinarchaeal proteins alongside other archaeal and bacterial FtsZs highlights their shared ancestry while underscoring the diversity of FtsZ architectures and interaction mechanisms that exist across lineages. Our analysis of the two distinct FtsZ proteins in Odinarchaeota reveals the unique filament assembly characteristics of FtsZ1 and FtsZ2. OdinFtsZ1 showed canonical protofilaments as reported for many bacterial FtsZs, suggesting a conserved mode of assembly across kingdoms (Fig. 2A,C). On the other hand, OdinFtsZ2 assembled into stacked ring-like structures, which have a striking resemblance to filament morphologies adopted by OdinTubulin (Akıl et al, 2022) (Fig. 2F,G). Higher resolution information of the spiral ring-like structures is necessary to comment whether the monomer interfaces within this assembly form a canonical protofilament interaction with a higher curvature. Since OdinFtsZ2 and OdinTubulin share a low similarity in terms of sequence and structure, similar filament morphologies may indicate their involvement in a common process or function. The observed difference in the filament morphologies of the two FtsZ proteins suggests probable functional specialization within archaeal cells, while their in vitro interaction hints at a potential collaborative role in cellular processes. We also note the similarity of the OdinFtsZ2 spiral ring-like assembly to the recently reported FtsA rings that prevent FtsZ filaments from bundling and align with the cryo-EM structures of FtsZ-FtsA filaments reconstituted within liposomes (Szwedziak et al, 2012; Krupka et al, 2017). This similarity may hint at a similar pairing between OdinFtsZ1 and OdinFtsZ2, which remains to be explored.

Our study also addresses the distinct membrane tethering mechanisms adopted by the dual Odin FtsZs. Interestingly, OdinFtsZ1 possesses an N-terminal amphipathic helix capable of membrane binding. Similar membrane binding has been found in different *Mycoplasma* clades where FtsZ proteins can interact with the membrane via either N- or C-terminal ends (Dutta et al, 2025). A comparable mechanism has also been reported in chloroplast FtsZ1 from *Arabidopsis thaliana* (An et al, 2024; Liu et al, 2022) where the C- terminal plays role in membrane binding. However, OdinFtsZ2, does not possess direct membrane binding but can indirectly bind via its membrane anchor, OdinSepF. SepF is highly conserved among archaea as a membrane anchor for FtsZ and co-exists in all archaea with FtsZ genes (Pende et al, 2021). It is interesting to observe that, unlike the bacterial homologs, the

archaeal SepF does not form higher-order assemblies, however, it interacts with OdinFtsZ2, which has a stacked ring-like assembly. OdinFtsZ2-SepF interaction is mediated by the C-terminal domain of OdinFtsZ2, a feature that seems to be present across different domains of life (Cendrowicz et al, 2012; An et al, 2024). Further, the observed interaction between FtsZ2 and SepF is consistent with previous findings in *H. volcanii*, another archaeon that encodes two FtsZ paralogs (Liao et al, 2021; Nußbaum et al, 2021). In contrast, in archaea with a single FtsZ homolog, such as *Methanobrevibacter smithii* (Pende et al, 2021), the sole FtsZ protein interacts with the membrane anchor. These observations suggest that when only one FtsZ is present, it directly associates with the membrane adaptor. However, in organisms possessing two FtsZ paralogs—such as Odinarchaeota and *H. volcanii*—adaptor-mediated membrane tethering appears to be selectively mediated by FtsZ2, indicating a division of function between the two paralogs.

The differentiation into the two modes of membrane attachment appears to be mediated by the linker length of the amphipathic helix and the SepF-binding motif to the globular domain in the respective sequences. N-terminal extensions to the globular fold of FtsZ2 sequences tend to be shorter compared to FtsZ1 across archaeal species, which may hinder a direct membrane attachment of FtsZ2 via its N-terminal region. However, the presence of longer C-terminal extensions of FtsZ2 compared to FtsZ1 could facilitate binding to SepF and thus to the membrane. The basis of retaining the redundant mechanism for membrane attachment of FtsZ remains unclear.

These observations add to the known spectrum of FtsZ behavior and offer comparative insights into the range of cytoskeletal organization strategies. Future studies should reveal more insights into the cellular functions of the distinct FtsZs in Odinarchaeota. It is possible to have a similar mechanism like in the archaeon *H. volcanii* (Liao et al, 2021), the two Odin FtsZs with distinct properties might cooperate in cell division. The conserved interaction between FtsZ2 and SepF further supports the possibility of a similar mechanism. Future studies involving the expression of OdinFtsZ proteins in *H. volcanii*—complementation assays in FtsZ deletion strains, localization studies and interaction analyses can reveal whether Odin FtsZs integrate into the division machinery and mimic native function as *H. volcanii* FtsZ proteins. Interestingly, the two OdinFtsZ genes are located alongside genes encoding ribosomal proteins and tRNA synthetases (Tamarit et al, 2022), suggesting a potential coordination between cellular biosynthesis and cell division cycles, often observed across species. Alternatively, we imagine that the two FtsZs might have distinct functions. OdinFtsZ1 filaments may adopt a classical Z-ring architecture for cell division, and OdinFtsZ2 (like CetZ) may play a role in functions such as membrane deformation for forming of intra-membrane structures or cell shape maintenance, since it shares an extruding helical loop in the C-terminal domain, like CetZ proteins (Appendix Fig. S1A). It is also speculated that the possible cell enlargement and membrane complexity during eukaryogenesis may have necessitated the emergence of a more rigid filamentous structure and led to the evolution of microtubule-like structures (Akıl et al, 2022). The stacked ring-like structures of OdinFtsZ2, like OdinTubulin as well, might be a step in generation of a multi-protofilament tubular cytoskeleton structures. Structural characterization of the stacked rings might provide interesting insights into the evolution of tubular structures.

Future structural and biochemical studies on synthetic chimeras of Odin FtsZs and OdinTubulin should shed light on the evolution and stabilization of straight protofilaments of FtsZ/tubulin into tubules. An understanding of the cellular functions of these proteins is largely hindered due to the lack of available culturing techniques for Asgard archaea. However, recent studies on the isolation and the culture of members of Lokiarchaeota (Imachi et al, 2020; Rodrigues-Oliveira et al, 2023) and of Hodarchaeales (proposed to be FECA's (First Eukaryotic Common Ancestor) closest relative) (Eme et al, 2023; preprint: Imachi et al, 2025) have opened new avenues for understanding the cellular roles of these Asgard archaeal proteins (Wollweber et al, 2025). Thus, future developments in culturing methods, advanced sequencing methods and genetic tools will help build a better understanding of the cell biology of the Asgard archaeal phylum with respect to the role of the multiple FtsZs.

# Methods

## Phylogenetic analysis

The maximum likelihood tree was generated for FtsZ using the protein sequences from Bacteria and Archaea. The phylogenetic tree was generated with IQTREE (version 1.6.12), and LG+G4+I was identified as the best-fit model by ModelFinder based on the BIC values (Trifinopoulos et al, 2016; Kalyaanamoorthy et al, 2017) (Fig. EV1). SepF tree was constructed using archaeal SepF sequences with LG + G4 + C60 as the best-fit model (Fig. EV5). All protein sequences were retrieved from the NCBI protein database, with a cutoff date 15 March 2025. For bacterial and plastid sequences, a direct keyword-based search was used. Archaea sequences were obtained using OdinFtsZ1 and OdinFtsZ2 as BLASTp queries against individual archaeal phyla, including MAGs (particularly from Asgard and TACK lineages), and through targeted retrieval from NCBI protein database. For SepF, sequences were compiled through direct database search to represent a broad sampling across archaea. To avoid redundancy, a single representative per species or strain was included where possible,

### Reagents and tools table

| Reagent/resource | Reference or source | Identifier or catalog number |
|---|---|---|
| **Experimental models** | | |
| **Recombinant DNA** | | |
| pET28a-6His-bdSUMO | Hatano et al, 2022 | |
| OdinFtsZ1 gene Fragment | GenBank: WEU40812.1 | Synthesized by Twist Bioscience |
| OdinFtsZ2 gene Fragment | GenBank: WEU40955.1 | Synthesized by Twist Bioscience |
| OdinSepF gene Fragment | GenBank: WEU40786.1 | Synthesized by Twist Bioscience |
| **Antibodies** | | |
| **Oligonucleotides and other sequence-based reagents** | | |
| pET28a-6his-bdSUMO-OdinFtsZ1-Fw | TGCTGCACCAAACTGGAGGCatgatctacatgcgtacgctt | |
| pET28a-6his-bdSUMO-OdinFtsZ1-Rv | AGTGCGGCCGCAAGCTTGttacagttcatcaagcgccagt | |
| pET28a-6his-bdSUMO-OdinFtsZ2-Fw | aTGCTGCACCAAACTGGAGGCatgcgcagtctgatcgaggatg | |
| pET28a-6his-bdSUMO-OdinFtsZ2-Rv | GTGCGGCCGCAAGCTTGttactaatacggtccagcccca | |
| pET28a-6his-bdSUMO-OdinFtsZ1(14-363)-Fw | gccaTGCTGCACCAAACTGGAGGCatgaaggcgcgcgtagagg | |
| pET28a-6his-bdSUMO-OdinFtsZ2-(1-355)-Rv | gctCGAGTGCGGCCGCAAGCTTGttaactcttcaccccagtaagaatc | |
| pET28a-6his-bdSUMO-Fw-OdinSepF | gccaTGCTGCACCAAACTGGAGGCATGGGA TTACTATCAAAAATATTTAGAAGGAAGAAGATC | |
| pET28a-6his-bdSUMO-Rv-OdinSepF | gtgctCGAGTGCGGCCGCAAGCTTGTT AACCGGTTTTACGCCAAATCTTTACA | |
| **Chemicals, enzymes, and other reagents** | | |
| Hi-fi DNA assembly master mix | New England BioLabs | Cat. no.: E2621L |
| HisTrap HP; 5 ml | Cytiva | Cat. no.: 17524701 |
| Cytiva Mono Q column (Mono Q™ 4.6/100 PE, Mono Q™ 10/100 GL) | Cytiva | Code. no.: 17-5179-01 |
| DOPG | Avanti Polar Lipids | Cat. no.: 840475C |
| Uranyl acetate | Ted Pella | Cat. no.: 19481 |
| GTP | Sigma-Aldrich | Cat. no.: G8877 |
| GTPγS | Sigma-Aldrich | Cat. no.: G8634 |
| GMPCPP | Jena Bioscience | Cat. no.: NU-405S |

| Reagent/resource | Reference or source | Identifier or catalog number |
|---|---|---|
| GDP | Sigma-Aldrich | Cat. no.: G7127 |
| Vitrobot filter paper | Ted Pella | Cat. no.: 47000-100 |
| Whatman Filter paper | Whatman | Cat. no.: 1001-110 |
| Quantifoil R1.2/1.3 Au 300 mesh grid | Quantifoil | Cat. no.: N1-C14nAu30-01 |
| Copper-Rhodium grid (3 mm) | Maxtaform | Cat. no.: M400-CR |
| **Software** | | |
| AlphaFold | https://alphafoldserver.com | |
| GraphPad Prism10 | https://www.graphpad.com | |
| Fiji (ImageJ2) | Schindelin et al, 2012 | |
| ChimeraX | https://www.cgl.ucsf.edu/chimerax/ | |
| BioRender | https://www.biorender.com | |
| HeliQuest | https://heliquest.ipmc.cnrs.fr/index.html | |
| EPU | Thermo Fisher Scientific | |
| cryoSPARC | https://guide.cryosparc.com/setup-configuration-and-management/how-to-download-install-and-configure | |
| RELION | https://github.com/3dem/relion | |
| Phenix | https://phenix-online.org/ | |
| Coot | https://www2.mrc-lmb.cam.ac.uk/personal/pemsley/coot/binaries/release/ | |
| PyMol | https://pymol.org/ | |
| Servalcat | https://github.com/keitaroyam/servalcat | |
| Weblogo | https://weblogo.berkeley.edu/logo.cgi | |
| MEGA7 | https://www.megasoftware.net | |
| **Other** | | |

prioritizing phylogenetic breadth. The sequences were aligned with MUSCLE and trimmed using MEGA7 based on alignment quality and consistency (Kumar et al, 2016). The support values for each clade, shown at the node, are inferred from ML IQTREE 1000 ultrafast bootstrap values. This dataset was assembled to maximize archaeal diversity rather than systematic screening, as the objective was to compare representative FtsZ homologs across taxonomic lineages.

## Plasmid construction

DNA fragments OdinFtsZ1 (UniProt Accession: A0A1Q9N645, OdinFtsZ2 (UniProt Accession: A0A1Q9N6K6) and OdinSepF (UniProt Accession: A0AAF0D343) were synthesized by Twist Biosciences. DNA fragments with the vector overhangs were amplified using PCR and gel extracted. Fragments and cut vectors were assembled using Hi-fi DNA assembly master mix (NEB) (New England BioLabs; cat. no: E2621L). The reaction mixture was transformed into DH5-alpha cells, and plasmids were isolated, confirmed and sequenced. All the plasmids used in this study are mentioned in the Appendix Table S1.

## Protein expression and purification

Proteins were expressed in BL21(DE3) cells grown in LB media containing 50 μg/mL of kanamycin at $OD_{600}$ 0.6 with 0.5 mM IPTG (isopropyl β-D-1-thiogalactopyranoside) for 4 h at 30 °C post-induction. Cells were harvested by centrifugation for 15 min at $5000 \times g$ at 4 °C. Cell pellets were lysed using an ultrasonic cell disruptor (Sonics) (Lysis Buffer—50 mM Tris pH 8, 200 mM NaCl, 10% glycerol) followed by clarification by centrifugation at $22,000 \times g$ at 4 °C. The supernatant was passed through a 5 ml HisTrap column (Cytiva HisTrap HP; 5 ml) pre-equilibrated with a binding buffer (50 mM Tris pH 8, 200 mM NaCl, 10 mM imidazole). Protein was eluted with a stepwise increase in concentration of imidazole. Elution fractions were loaded onto 12% SDS-PAGE gel, and pure fractions were pooled. Fractions were subjected to cleavage with Ulp protease (1:100 moles ratio of protease: protein) for 2 h (overnight cleavage for OdinFtsZ1). The protein buffer was then exchanged with an imidazole-free buffer (50 mM Tris pH 8, 200 mM NaCl). This was then passed through the Ni-NTA column, pre-equilibrated with the same buffer to separate the cleaved SUMO tag and the un-cleaved protein. Pure protein without the tag

was collected in flow-through, and then the buffer was exchanged with storage buffer (50 mM HEPES pH 7.4, 50 mM KCl). Cytiva Mono Q column (Mono Q™ 4.6/100 PE, Mono Q™ 10/100 GL) was used for ion exchange chromatography. In total, 50 mM HEPES pH 7.4, 50 mM KCl served as the binding buffer for Mono Q and the protein was eluted using a linear gradient of 0 to 40% of elution buffer (50 mM HEPES pH 7.4, 1 M KCl). After ion exchange chromatography, required fractions were pooled and dialyzed against the storage buffer with a final KCl concentration of 50 mM. This protein was then concentrated using Centricon (10 kDa), flash-frozen and stored at −80 °C.

## Light scattering assay

To monitor the polymerization of OdinFtsZ1 and OdinFtsZ2, 90° angle light scattering (Horiba Fluoromax-4) was used. The 200 µL reaction mix contained the buffer (50 mM HEPES, pH 7.4, 50 mM KCl), 2 mM GTP, 5 mM Mg$^{2+}$ and protein at given concentrations. The emission and excitation wavelengths for the assay was set to 350 nm, and the slit width used was 1.5 nm. The reaction mix (without GTP) was added to a cuvette with a 1 cm path length and was placed into a temperature-controlled cuvette chamber set at 30 °C. Prior to GTP addition, reading was taken for setting a baseline. After 300 s, GTP was added to the cuvette to achieve a final concentration of 2 mM, and readings were recorded for 40 min. The relative intensity profile was obtained by dividing all the individual intensity values by the average baseline intensity obtained by averaging intensities from 1 to 300 s (Fig. 1C,D; Appendix Fig. S2B,C).

## Pelleting assay

Purified proteins OdinFtsZ1 and OdinFtsZ2 were spun at 22,000×$g$ for 20 min at 4 °C initially to remove any aggregates. The protein concentration of the supernatant was then estimated using the Bradford assay (Bradford, 1976). The reaction volume for the pelleting assay was 50 µL, which consisted of 10 µM of protein, 3 mM GTP, 5 mM Mg$^{2+}$ in the assay buffer (50 mM HEPES, pH 7.4, 50 mM KCl). Presence of the protein in the pellet fraction upon high-speed ultracentrifugation was monitored in the absence and presence of GTP. Also, both proteins were added in equimolar concentration (10 µM) to monitor co-pelleting. Once the reactions were prepared in the ultracentrifuge (Beckman Coulter Optima MAX-XP Ultracentrifuge) tubes, they were kept inside the rotor (TLA 120.2) set at 30 °C for 5 min and followed by centrifugation at 100,000 × $g$ for 15 min. After the spin, the supernatant was transferred to a fresh vial and the pellet was washed gently with 100 µL buffer (50 mM HEPES pH 7.4, 50 mM KCl). The pellet was resuspended in 50 µL of 5× SDS-PAGE running buffer (125 mM Tris base, 962 mM glycine and 1.73 mM SDS). Equal volumes of supernatant and pellet fractions were mixed with 2× Laemmli buffer and loaded onto a 12% SDS-PAGE gel to visualize the proteins (Fig. EV4A,B). Protein band intensities were analyzed using Fiji (ImageJ2) analysis software (Schindelin et al, 2012). The intensity of bands was calculated as the difference in intensity of a band from that of the average intensity of the background. The percentage of protein in the pellet fraction was calculated by dividing the intensity of the pellet fraction by the sum of the intensities of the pellet and supernatant fractions.

## Pull-down assay for FtsZ

To test the interaction between 6His-SUMO tagged OdinFtsZ with OdinSepF, purified OdinFtsZ1 and OdinFtsZ2 were kept bound to Ni-NTA resin (Appendix Fig. S3C) and incubated with untagged OdinSepF. The total reaction volume was 500 µL of binding buffer (50 mM Tris pH 8, 200 mM NaCl) with equimolar amounts of both the proteins (resin-bound protein and untagged OdinSepF), incubated at 4 °C for 1 h. After incubation, the beads were washed (thrice) with a wash buffer (50 mM Tris pH 8, 200 mM NaCl, 25 mM imidazole) to remove non-specific binding. Proteins bound with Ni-NTA resin were then eluted with a 500 µL elution buffer (50 mM Tris pH 8, 200 mM NaCl, 250 mM imidazole). A control experiment with Ni-NTA resin-bound 6His-SUMO was done to eliminate possible SUMO-based binding. Each fraction (Total, Un-bound and elute) was collected and loaded on 15% SDS-PAGE gel to visualize the proteins (Fig. 3C,D).

## Liposome preparation

Lipids utilized in all the liposome-based experiments, namely, dioleoyl phosphatidylglycerol (DOPG; 840475C) was acquired from Avanti Polar Lipids. To make a 2 mM stock solution of lipids, the required amount of lipid (100% DOPG) in chloroform was aliquoted in a small test tube and then dried with nitrogen (or argon) to remove chloroform. After this, the dried lipids were resuspended in HK50 buffer (50 mM HEPES pH 7.4, 50 mM KCl), such that the final stock concentration was 2 mM. This lipid solution was then extruded through a 100 nm polycarbonate membrane (Avanti Polar Lipids) and used for all the liposome-based assays.

## Liposome pelleting assay

Purified proteins were spun at 100,000 × $g$ for 15 min at 4 °C to remove any aggregates. The concentration of supernatant was then estimated using the Bradford assay. The reaction volume used for the assay was 100 µL, which consisted of 2 µM protein, varying concentrations of liposome (0–600 µM) in HK50 buffer. In total, 100 µL reaction was made in an ultracentrifuge tube and incubated for 15 min at 30 °C. Then, the reactions were spun at 100,000 × $g$ for 15 min. After the spin, the supernatant was transferred to a fresh vial and the pellet was washed gently with 200 µL HK50 buffer and resuspended in 50 µL of the same buffer. Equal volumes of supernatant and pellet fractions were mixed with 2× Laemmli buffer and loaded onto a 12% SDS-PAGE gel to visualize the proteins. Protein band intensities were analyzed using Fiji (ImageJ2) (Schindelin et al, 2012). The intensity of bands was calculated as the difference in intensity of a band from that of the average intensity of the background. The percentage of protein in the pellet fraction was calculated by dividing the intensity of the pellet fraction by the sum of the intensities of the pellet and supernatant fractions.

## Structural analysis

All the AlphaFold-predicted structures and crystal structures were downloaded from UniProt for structural analysis. Superimposition of structures and the rest of the structural analysis were done using

ChimeraX (Goddard et al, 2018). Sequence conservation was assessed using sequence logo generated from the multiple sequence alignment of all sequences used in the phylogenetic analysis, created with Weblogo (Fig. 1B; Appendix Fig. S3D). The AlphaFold structure prediction (Fig. 1A) and the sequence logo (Fig. 1B; Appendix Fig. S3D) were generated based on the sequences of OdinFtsZ1 (UniProt Accession: A0A1Q9N645, now updated to A0AAF0ID74) and OdinFtsZ2 (UniProt Accession: A0A1Q9N6K6, now updated to A0AAF0D3V8). HeliQuest (Gautier et al, 2008) was used for amphipathic helix prediction, helical wheel representation and calculation of hydrophobic moment and hydrophobicity parameters (Appendix Fig. S3A; Fig. 4A).

## Size-exclusion chromatography coupled with multi-angle light scattering (SEC-MALS)

SEC-MALS experiment enabled the accurate mass estimation for purified OdinSepF. Superdex 75 Increase 10/300 GL column was used for SEC, which was connected to an Agilent HPLC unit with an 18-angle light scattering detector (Wyatt Dawn HELIOS II) and a refractive index detector (Wyatt Optilab T-rEX). The experiments were performed at room temperature. The column was equilibrated with HK50 buffer at a flow rate of 0.4 ml/min. BSA at 2 mg/mL was used to calibrate the system. Purified OdinSepF (2 mg/ml, 100 μl) was subsequently loaded to estimate the molecular weight of the eluted peaks. The Zimm model implemented in ASTRA software was used for the curve fitting and estimation of molecular weights. GraphPad Prism was used to plot the average molar mass from fitted plots (Appendix Fig. S3B).

## Negative staining electron microscopy

The protein samples were thawed, diluted in 50 mM HEPES pH 7.4, 50 mM KCl, 5 mM $MgCl_2$ and incubated with a final concentration of 2 mM nucleotide (GTP, GTPγS, GDP or GMPCPP) at 30 °C for 15 min or as indicated. The final concentration of OdinFtsZ1 was 0.04 mg/mL, and that of OdinFtsZ2 was 0.1 mg/mL for all conditions. Maxtaform Cu/Rh 3-mm grids were coated with homemade carbon film (Edwards BT150) on one side. The carbon-coated side was glow-discharged in a Quorum GloQube glow discharge instrument at 20 mA for 10 s. In total, 3 μL of the protein sample was applied to the grid, incubated for a minute, and washed three times with MilliQ water. The excess water was blotted with Whatman filter paper No. 1 and stained with 2% uranyl acetate twice with a final incubation for 1 min, after which the excess stain was removed and air dried for 1 min. The grids were then imaged using Talos F200S G2 TEM operated at 200 kV and a CETA camera.

## Cryo-EM Sample preparation, data collection, and image processing

**For OdinFtsZ1**, a solution containing 0.4 mg/ml FtsZ1, 50 mM HEPES pH 7.4, 50 mM KCl, 5 mM $MgCl_2$, and 2 mM GMPCPP was incubated on ice for 30 min. Quantifoil grids (R1.2/1.3 Au 300 mesh) were glow-discharged using the PELCO easiGlow™ system at 25 mA for 60 s. 3 μl of the protein solution was applied to the glow-discharged grids in a Vitrobot Mark IV chamber (Thermo Fisher Scientific, USA) equilibrated at 16 °C and 100% humidity. The grids

were blotted with a blot force of 0 for 3.5 s, plunge-frozen in liquid ethane, and subsequently transferred to liquid nitrogen for storage prior to imaging on the Krios G3i microscope at the National Electron cryoMicroscopy Facility, BLiSC Campus, Bangalore. Grids were screened for optimal ice thickness and particle distribution.

Two datasets were collected at 0° and 20° stage tilt, respectively, at a nominal magnification of 75,000x, corresponding to a pixel size of 1.07 Å, using a Falcon 3 detector in counting mode. The set defocus values ranged between −1.8 μm and −3.0 μm. Movies were fractionated into 25 frames, with a total electron dose of 23.5 e⁻/Å² (0° tilt) and 23 e⁻/Å² (20° tilt).

Following data collection, images were processed in cryoSPARC (Punjani et al, 2017) version 4.5.3. The 0° dataset was initially acquired and processed till 3D reconstruction and helical refinement step. However, the resulting density map showed significant anisotropy, and the estimated helical twist (~2°) suggested a near-linear arrangement of subunits with maps lacking high-resolution features that were not reflective of the resolution obtained. This indicated a significant orientation bias, which was addressed by collecting an additional dataset at a 20° stage tilt to improve angular coverage.

Each dataset was processed separately up to particle extraction and 2D classification. In total, 2004 movies (0° tilt) and 1106 movies (20° tilt) were imported, motion-corrected, and used for CTF estimation. A filament tracer job was initiated with a filament diameter of 45 Å and a segment separation distance of 1 filament diameter. From these, 1,798,671 particles (0° tilt) and 978,423 particles (20° tilt) were extracted using a box size of 256 pixels. After two rounds of 2D classification, particles corresponding to single filaments showing visible secondary structural features were selected, yielding 372,802 particles (0° tilt) and 377,883 particles (20° tilt). The selected particles (750,685 in total) were merged for a final round of 2D classification, and 524,514 particles were selected for further processing. Six ab-initio models were generated, of which four classes displaying well-defined features comprising 356,932 particles were selected. These were subjected to helical refinement without specifying helical symmetry parameters (HS1). Following initial refinement, a symmetry search revealed a helical rise of 44.5 Å and a twist of 2.02°. Using these parameters, a second helical refinement was performed with helical symmetry imposed. After further global CTF refinement, a final helical refinement yielded a map at ~3.55 Å resolution (FSC = 0.143), with a maximum symmetry order of 4 during reconstruction.

Examination of the resulting sym_sharp map in ChimeraX (Goddard et al, 2018; Meng et al, 2023) revealed the map to be of the wrong hand. To correct this, the HS1 map was flipped in ChimeraX and re-imported into cryoSPARC using volume tools. A new helical refinement was then initiated with parameters: rise of 44.5 Å and twist of −2.02°, as recommended in the cryoSPARC tutorials (https://guide.cryosparc.com/processing-data/all-job-types-in-cryosparc/3d-refinement/job-homogeneous-reconstruction-only; https://guide.cryosparc.com/processing-data/all-job-types-in-cryosparc/helical-reconstruction-beta/helical-symmetry-in-cryosparc). After another round of global CTF refinement, the final helical refinement produced a map at ~3.57 Å resolution (FSC = 0.143), with optimized parameters of 44.5 Å rise, −2.7° twist, and a maximum symmetry order of 4 (Fig. EV2).

For model building, an AlphaFold-predicted model (UniProt Accession: A0AAF0ID74) of OdinFtsZ1 monomer (residues

**Table 1.** Data collection and processing for OdinFtsZ1.

| | OdinFtsZ1 | |
| --- | --- | --- |
| | 0° Tilt | 20° Tilt |
| Magnification (nominal) | 75,000× | 75,000× |
| Voltage (kV) | 300 | 300 |
| Electron exposure (e/ Å²) | 23.5 | 23 |
| Nominal defocus range (μm) | −1.8 to −3.0 | −1.8 to −3.0 |
| Pixel size (Å) | 1.07 | 1.07 |
| No. of imported movies | 2004 | 1106 |
| Symmetry imposed | C1 | |
| No. of particle images after 2D | 750,685 (372,802 + 377,883) | |
| No. of final particle images | 356,932 | |
| Map resolution at FSC 0.143 (Å) | 3.6 | |
| Map resolution range (Å) | 3.6–8.0 | |
| Map sharpening B factor (Å²) | −175.1 | |
| **Refinement** | | |
| Initial model used | AlphaFold generated | |
| **Model composition** | | |
| Non-hydrogen atoms | 8912 | |
| Protein residues | 1192 | |
| Ligands | 4 | |
| **B factors (Å²)** | | |
| Protein | 115.87 | |
| GMPCPP | 111.16 | |
| **R.M.S deviation** | | |
| Bond lengths (Å) | 0.004 | |
| Bond angles (°) | 0.573 | |
| **Validation** | | |
| Molprobity score | 2.51 | |
| Clashscore | 8.95 | |
| Poor rotamers (%) | 4.59 | |
| **Ramachandran plot** | | |
| Favored (%) | 94 | |
| Allowed (%) | 6 | |
| Disallowed (%) | 0 | |

34–332) was initially fitted into the refined sym_sharp map using ChimeraX. One monomer was used to examine the fit of the model to the density, and the model was adjusted manually in Coot (Emsley and Cowtan, 2004; Casañal et al, 2020), using KpFtsZ (PDB: 8ibn) as a reference. Real-space refinement of the model against the map was carried out using Phenix (Afonine et al, 2018; Liebschner et al, 2019) and the fit of the model into the map for two monomers in the filament is shown in (Fig. 2D). Analysis of the Servalcat difference density map revealed non-protein density at the subunit interface (Fig. 2E), which was subsequently modeled as GMPCPP (Yamashita et al, 2021), and the model was expanded to cover the 4 monomers in the filament. Although residues 34–332 have been modeled, additional density is visible at the N-

terminus, suggesting the presence of unresolved residues (Fig. EV3C). The map quality of the unmodeled density (red) is too poor to allow for reliable modeling. Data collection, processing, and refinement statistics are summarized in Table 1.

**For OdinFtsZ2**, a 4 mg/mL protein solution was incubated with 2 mM GTP at 30 °C for 15 min prior to grid freezing, following the same protocol used for OdinFtsZ1. However, 0.5% CHAPSO was added just before grid freezing to facilitate the visualization of side views. Data was collected at a nominal magnification of 75,000×, corresponding to a pixel size of 1.07 Å, using a Falcon 3 detector in counting mode. The set defocus values ranged between −1.8 μm and −3.0 μm. Movies were fractionated into 25 frames, with a total electron dose of ~24 e⁻/Å². Following data collection, images were processed in RELION 4.0(Scheres, 2012; Kimanius et al, 2021; Zivanov et al, 2018).

In total, 1879 movies were imported and the movie frames were aligned with the inbuilt MotionCorr routine in RELION, followed by CTF estimation with CTFFIND4 (Rohou and Grigorieff, 2015) and RELION 4.0 (Kimanius et al, 2021). For OdinFtsZ2, particles were manually picked, resulting in 143,398 particles, which were extracted with a box size of 540 pixels. After 2D classification, 31,538 particles with only two stacks and the top and side views were selected. However, the resolution of the class averages was poor. Attempts at 3D reconstruction using various symmetry options (C1, C2, C24–C27) did not result in a high-resolution map. This is likely due to the variability in the number of subunits within the doublet, which prevented the application of global symmetry. Larger rings (larger than doublet) were frequently observed, but due to the variability in the number of rings, they were not included in the class averaging.

## Data availability

Atomic models and cryo-EM maps of OdinFtsZ1 have been deposited in the Protein Data Bank (PDB, https://www.rcsb.org) and Electron Microscopy Data Bank (EMDB, https://www.ebi.ac.uk/emdb) under accession codes PDB 9V7V (https://www.wwpdb.org/pdb?id=pdb_00009v7v) and EMD-64825 (https://www.ebi.ac.uk/emdb/EMD-64825), respectively. The representative cryo-EM images shown in Fig. 2 are provided in the source data file.

The source data of this paper are collected in the following database record: biostudies:S-SCDT-10_1038-S44318-025-00529-7.

## Peer review information

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

## Acknowledgements

We thank the EM facilities at the Division of Biological Sciences, Indian Institute of Science, and the Electron Microscopy facility at NCBS, Bengaluru and Electron microscopy facility at the Department of Physics, IISER Pune, Jyotsna Singh and Radha Chouhan, NCCS Pune for SEC-MALS facility and Sivang Goswami, IISER Pune, for facilitating the SEC-MALS experiment. All the cryo-EM data were collected at the National Electron cryoMicroscopy Facility at the BLiSC campus in Bangalore. AD and JK acknowledge the fellowship from IISc. AK acknowledges the PhD fellowship from the Department of Biotechnology. SR acknowledges a fellowship from DAE. SD acknowledges the CSIR for the SRF fellowship. We thank Prof. Buzz Baum, Dr. Alex Bisson and Prof. PN Rangarajan for their feedback. A schematic diagram is created using BioRender. SP acknowledges the Department of Biotechnology-Wellcome Trust India Alliance intermediate fellowship (IA/I/21/1/505633), SERB SRG grant (SRG/2021/001600) and an Indian Institute of Science (IISc) start-up

grant. PG acknowledges support from the Department of Biotechnology-Wellcome Trust India Alliance Senior Fellowship (IA/S/23/1/506755) and Ministry of Education STARS grant (STARS2/2023-1015). RS acknowledges research grant support from the Department of Atomic Energy (DAE) and the Department of Biotechnology (BT/INF/22/SP33046/2019). The National cryo-EM facility and the computing cluster were supported by the Department of Biotechnology B-Life grant DBT/PR12422/MED/31/287/2014, and research in the KRV lab is supported by the Department of Atomic Energy, Government of India, Project Identification No. RTI 4006. SRP acknowledges the Department of Biotechnology-Wellcome Trust India Alliance intermediate fellowship (IA/I/20/1/504921) and the Max Planck Partner Group grant.

## Author contributions

**Jayanti Kumari**: Resources; Data curation; Formal analysis; Validation; Investigation; Visualization; Methodology; Writing—original draft; Writing—review and editing. **Akhilesh Uthaman**: Data curation; Formal analysis; Validation; Investigation; Visualization; Methodology; Writing—review and editing. **Sucharita Bose**: Data curation; Formal analysis; Validation; Investigation; Visualization; Methodology; Writing—review and editing. **Ananya Kundu**: Data curation; Formal analysis; Validation; Visualization; Writing—review and editing. **Vaibhav Sharma**: Data curation; Formal analysis; Validation; Investigation; Methodology; Writing—review and editing. **Soumyajit Dutta**: Data curation; Formal analysis; Investigation; Visualization; Writing—review and editing. **Anubhav Dhar**: Resources; Writing—review and editing. **Srijita Roy**: Resources; Writing—review and editing. **Ramanujam Srinivasan**: Resources; Supervision; Funding acquisition; Writing—review and editing. **Samay Pande**: Formal analysis; Supervision; Funding acquisition; Validation; Investigation; Methodology; Writing—review and editing. **Kutti R Vinothkumar**: Formal analysis; Supervision; Funding acquisition; Validation; Investigation; Visualization; Methodology; Writing—review and editing. **Pananghat Gayathri**: Formal analysis; Supervision; Funding acquisition; Validation; Investigation; Visualization; Methodology; Writing—review and editing. **Saravanan Palani**: Conceptualization; Resources; Formal analysis; Supervision; Funding acquisition; Validation; Investigation; Methodology; Writing—original draft; Project administration; Writing—review and editing.

Source data underlying figure panels in this paper may have individual authorship assigned. Where available, figure panel/source data authorship is listed in the following database record: biostudies:S-SCDT-10_1038-S44318-025-00529-7.

## Disclosure and competing interests statement

The authors declare no competing interests.

# Expanded View Figures

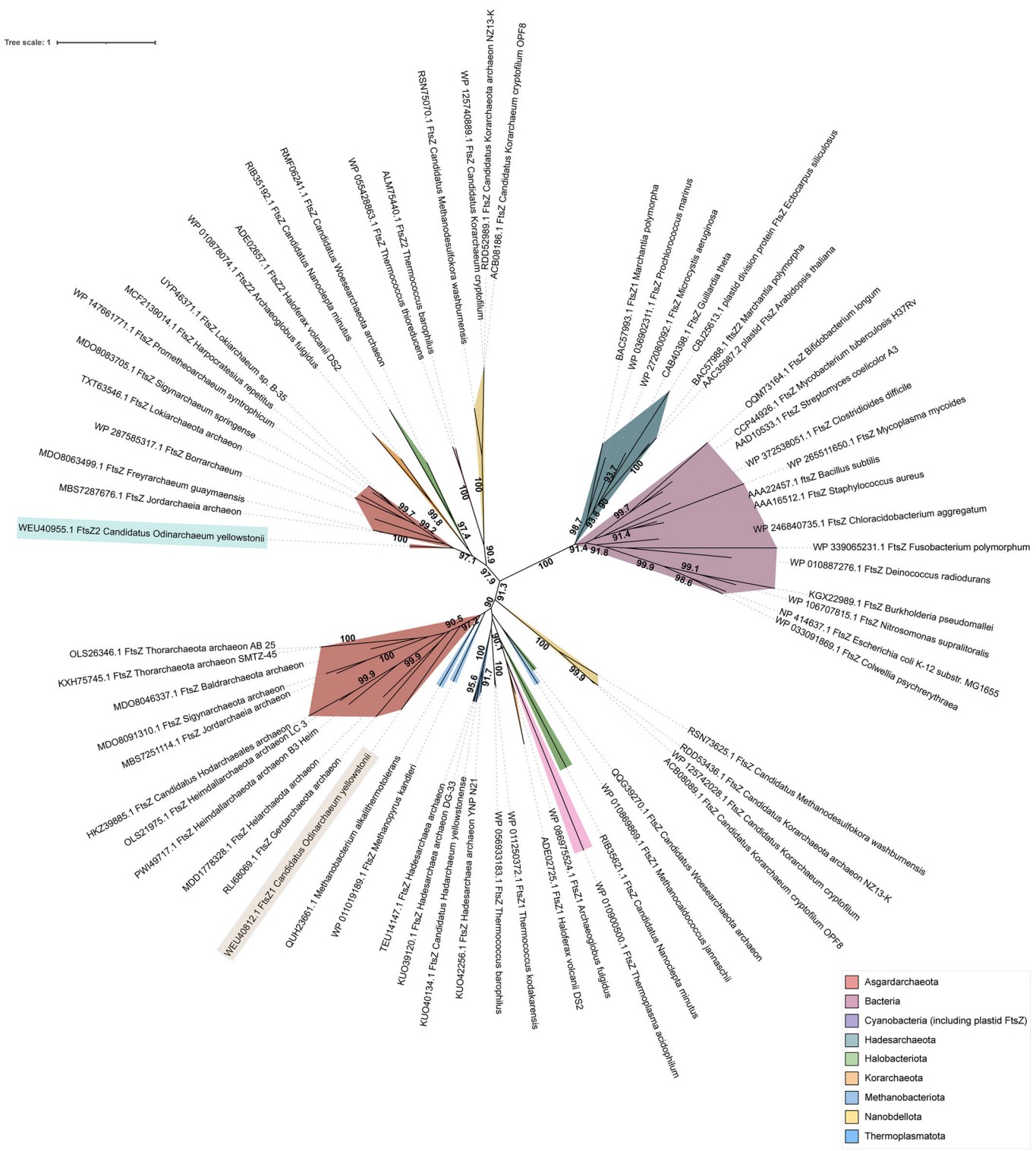

**Figure EV1.  Maximum likelihood (ML) tree of representative FtsZ sequences.**

The unrooted ML tree was generated for FtsZ using the protein sequences from bacteria and archaea. The phylogenetic tree was generated using IQTREE with model LG + G4 + I (69 sequences, 396 amino acid sites). Source data are available online for this figure.

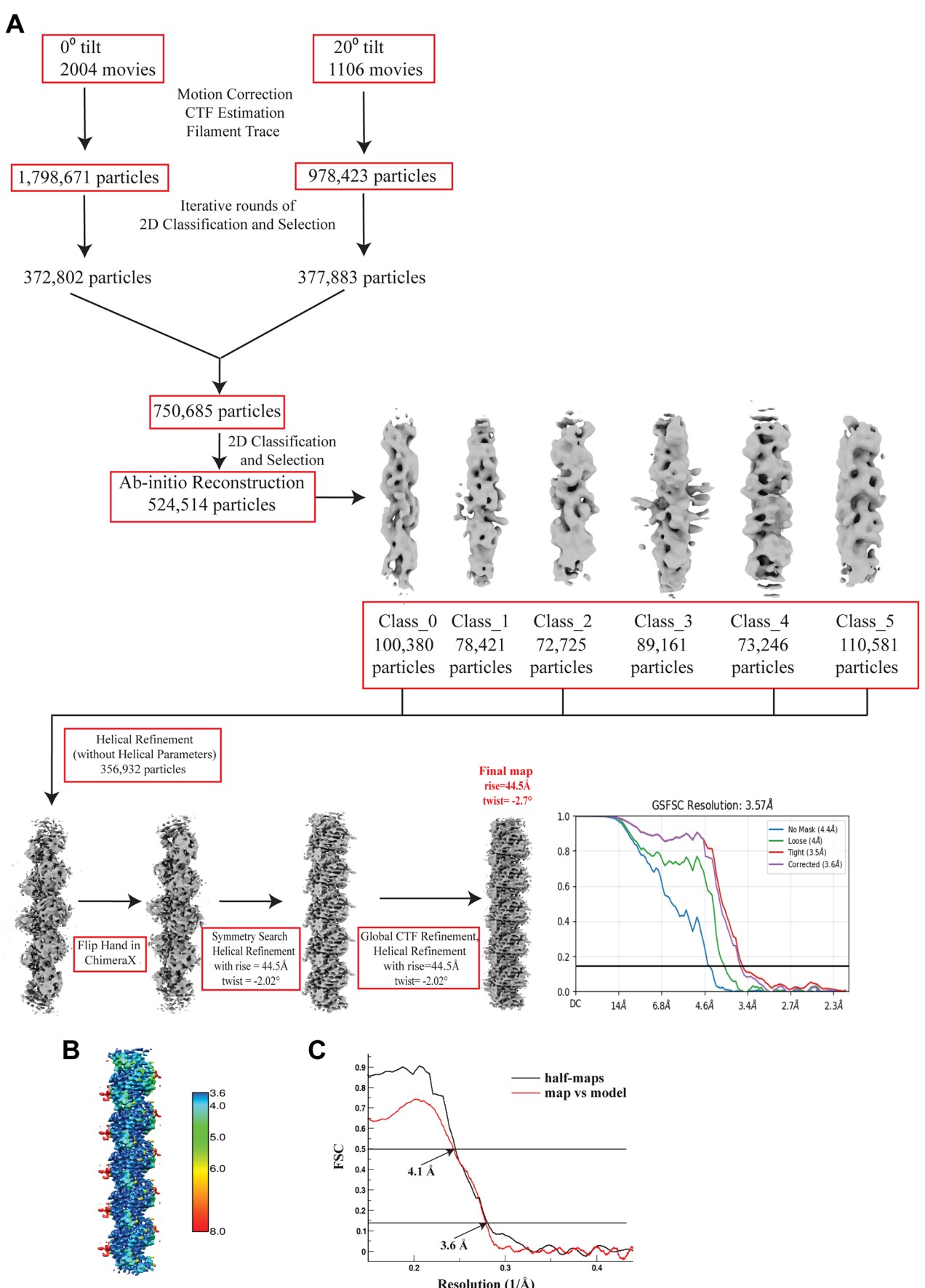

◀ **Figure EV2. Cryo-EM data processing workflow of OdinFtsZ1 single filament.**

(A) Two datasets were collected, untilted and another with a 20° tilt and processed individually till 2D classification and then the best classes were combined. These classes were processed in cryoSPARC, and the workflow is described. The GSFSC curves from cryoSPARC are shown with different masks and an estimated overall resolution of 3.6 Å. (B) The local resolution plot of the map shows the core of the filament resolved better and the periphery poorly resolved as expected. (C) Comparison of the FSC's derived from the half maps (@ 0.143, black curve) and map vs model (@ 0.5, red curve) indicates a resolution of 3.6 and 4.1 Å, respectively. Source data are available online for this figure.

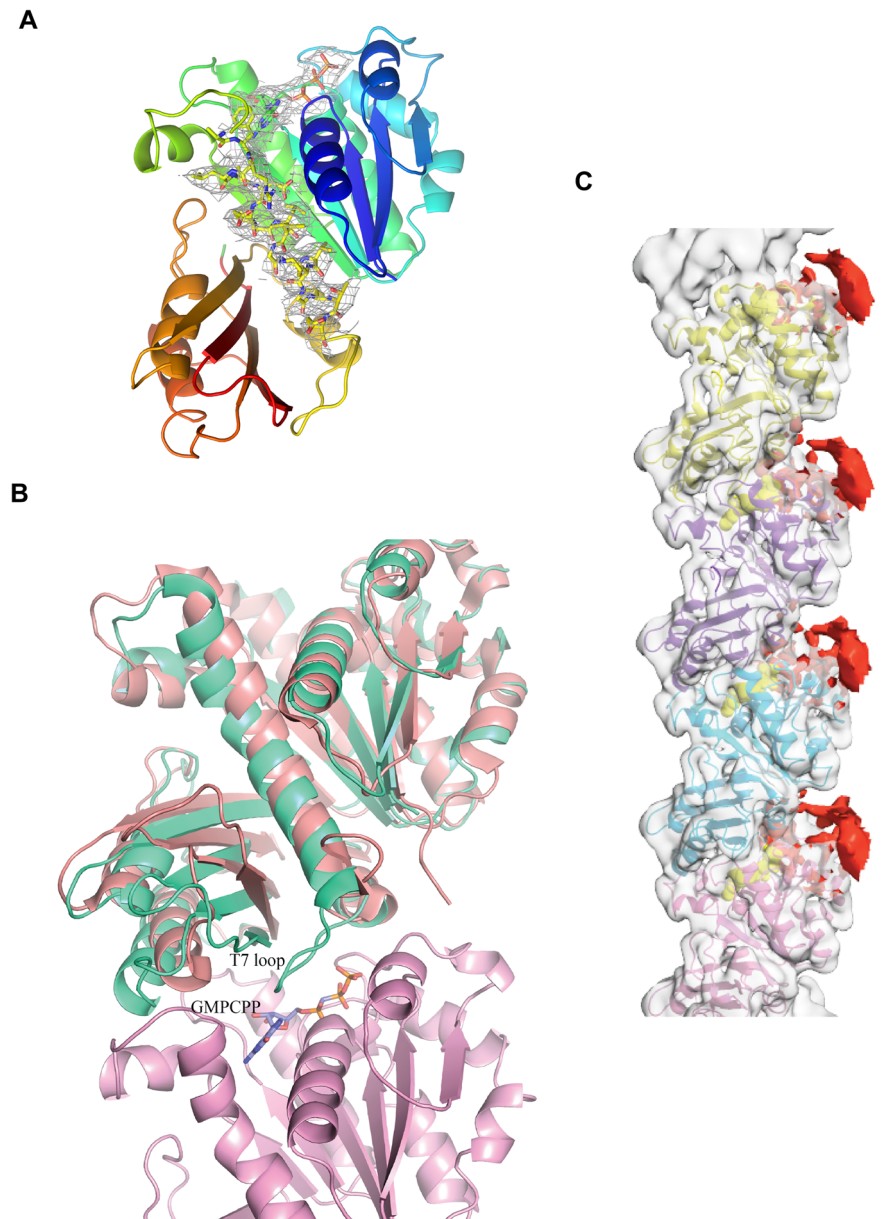

**Figure EV3.    OdinFtsZ1 model fit into the final sharpened map.**

(**A**) The OdinFtsZ1 monomer is shown in cartoon representation and colored in rainbow from the N-terminus (blue) to the C-terminus (red). The central helix (H7) and GMPCPP are represented as sticks and encased in the final_sym_sharpened map contoured at ~7σ in PyMol. (**B**) OdinFtsZ1 is present in T-conformation in the filaments. Superimposition of OdinFtsZ1 (monomer A, salmon) on *Staphylococcus aureus* (SaFtsZ) protein in R conformation (PDB ID: 5H5G, Chain B), (green) shows that the central helix is pushed one turn down with the catalytic T7 loop inserted into the subunit interface which is in close proximity to the GMPCPP molecule bound to the second monomer (monomer B, pink). (**C**) Unmodelled density at the N-terminus of the OdinFtsZ1 monomer. Although residues 34–332 have been modeled, additional density (red) is visible at the N-terminus, suggesting the presence of unresolved residues. The map quality in this region is low to allow for unambiguous modeling. Source data are available online for this figure.

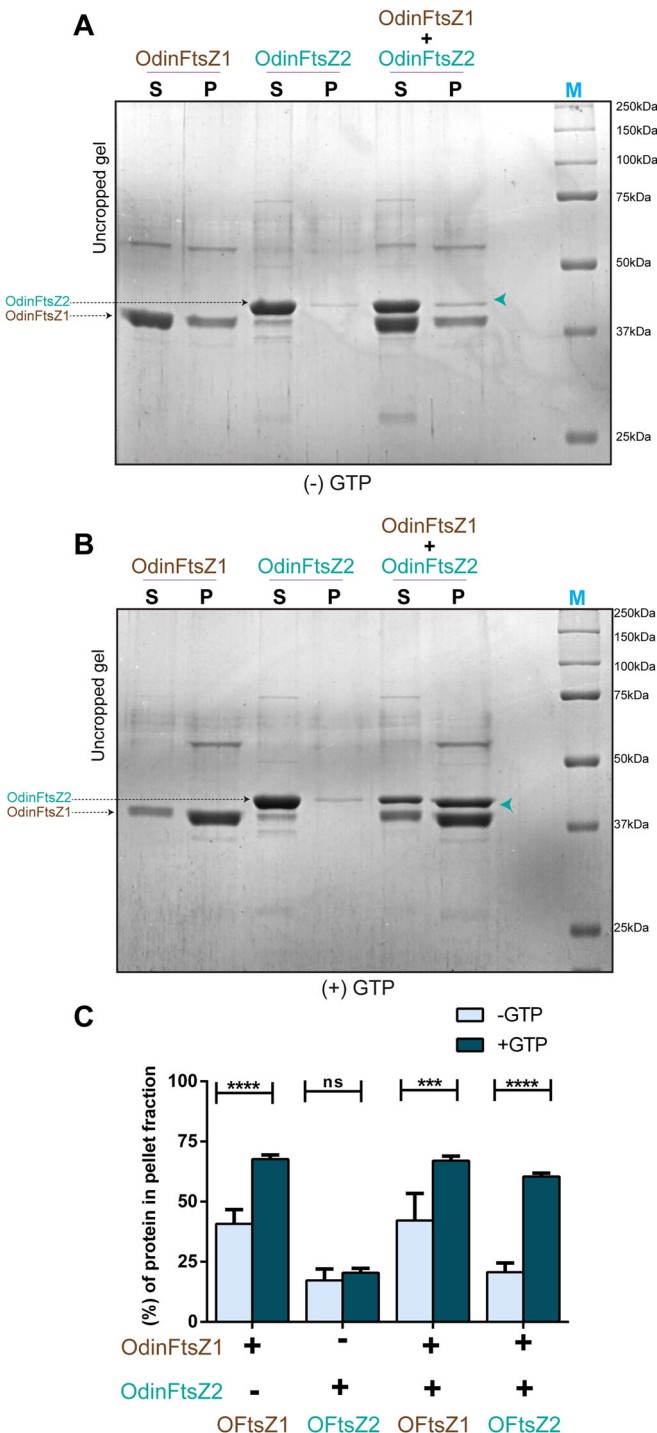

**Figure EV4. OdinFtsZ1 and OdinFtsZ2 co-pellet on ultracentrifugation.**

(A) Representative 12% SDS-PAGE gel for sedimentation of individual proteins OdinFtsZ1 (10 μM), OdinFtsZ2 (10 μM) and the co-sedimentation when incubated in equimolar concentration (10 μM) in the absence of GTP. (B) Representative 12% SDS-PAGE gel for sedimentation of individual proteins OdinFtsZ1 (10 μM), OdinFtsZ2 (10 μM) and the co-sedimentation when incubated in equimolar concentration (10 μM) in the presence of GTP nucleotide. Both supernatant (S) and pellet (P) fractions upon ultracentrifugation at 100,000×*g* were loaded. (C) Representative plot for the mean percentage of protein in the pellet fraction (*y* axis) with and without GTP. The pellet (or supernatant) fraction intensities corresponding to the band were calculated as the intensity of the pellet (or supernatant) divided by the sum of the two intensities and represented in the graph as a percentage. These percentages are represented as mean ± standard error of the mean (SEM) in the graph. '+' or '−' below the bars denote the presence/absence of Odin FtsZs, while the OFtsZ1/OFtsZ2 labeled below the bottom lines denote the protein band which has been quantified. (*N* = 3 (results replicated over three independent experiment), Two-way ANOVA with Sidak's multiple comparisons test was used, ns -non-significant, ***$P = 0.0001572$, ****$P = 0.0000638$ (for OdinFtsZ1), ****$P = 0.0000005$ (for OdinFtsZ2(+Z1))). Source data are available online for this figure.

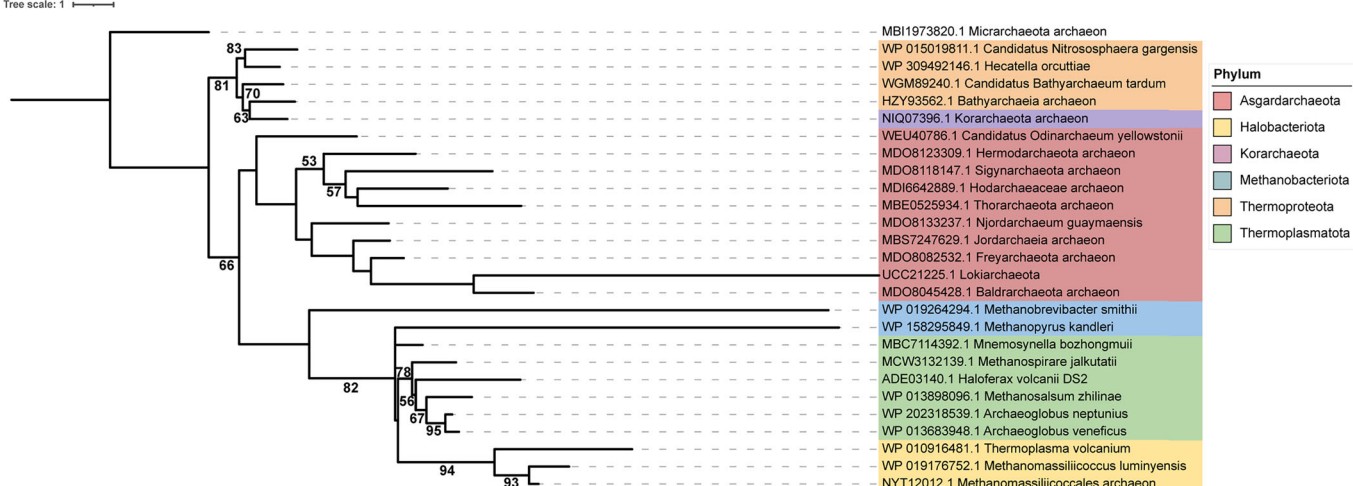

**Figure EV5. SepF as a universal archaeal membrane anchor protein.**

The ML tree for representatives archaeal SepF sequences, generated using IQTREE with LG + G4 + C60 model (27 sequences, 118 amino acid sites). The tree was rooted using Micrarchaeota (DPANN group) as an outgroup. Source data are available online for this figure.

