## [Peer Review File · The EMBO Journal]

Distinct filament morphology and membrane tethering features of the dual FtsZ paralogs in Odinarchaeota

Jayanti Kumari, Akhilesh Uthaman, Sucharita Bose, Ananya Kundu, Vaibhav Sharma, Soumyajit Dutta, Anubhav Dhar, Srijita Roy, Ramanujam Srinivasan, Samay Pande, Vinothkumar Kutti Rangunath, Pananghat Gayathri, and Saravanan Palani

Corresponding author(s): Saravanan Palani (spalani@iisc.ac.in) , Pananghat Gayathri (gayathri@iiserpune.ac.in)

Review Timeline:

Submission Date:	7th Feb 25
Editorial Decision:	12th Mar 25
Revision Received:	16th Jun 25
Editorial Decision:	14th Jul 25
Revision Received:	22nd Jul 25
Accepted:	25th Jul 25

Editor: Hartmut Vodermaier

Transaction Report:

Dr. Saravanan Palani
Indian Institute of Science
Biochemistry
Bangalore, Karnataka 560012
India

12th Mar 2025

Re: EMBOJ-2025-120418
Distinct filament morphology and membrane tethering features of the dual FtsZs in Odinarchaeota

Dear Saravanan,

Thank you again for submitting your study on dual FtsZs in Odinarchaeota to The EMBO Journal. It has now been assessed by four expert referees, whose comments are copied below. I am happy to say that all of them are in principle supportive of the study, and that we would therefore like to consider a revised version further for publication in our journal.

As you will see, each referee brings a number of constructive suggestions for improving the manuscript prior to publication. Referees 1 and 2 are mainly concerned about phylogenetic/taxonomy considerations and evolutionary interpretations and discussions, and referee 3 additionally raises a few technical issues. Furthermore, the reports recommend some additional computational/analytical mining of data, the most significant of which are referee 4's requests for more in-depth analyses of the structural data. In line with this, I feel that addressing the referees' well-taken and reasonable points should greatly further strengthen the study and its impact.

Please note that it is our policy to consider only a single round of major revision, making it important to fully respond to all comments at the time of resubmission; therefore, please do not hesitate to get back to me in case you would like to clarify/discuss any of the referees' points or plans for addressing them ahead of time. We would also be open to extending the revision deadline if that should be helpful.

Detailed information on preparing, formatting and uploading a revised manuscript can be found below and in our Guide to Authors, and adhering to them as closely as possible shall greatly facilitate editorial processing upon resubmission. Thank you again for the opportunity to consider this work for The EMBO Journal, and I look forward to your revision in due time.

With kind regards,

Hartmut

3) Revised manuscript text (including main tables, and figure legends for main and EV figures) has to be submitted as editable

text file (e.g., .docx format). We encourage highlighting of changes (e.g., via text color) for the referees' reference.

4) Each main and each Expanded View (EV) figure should be uploaded as individual production-quality files (preferably in .eps, .tif, .jpg formats). For suggestions on figure preparation/layout, please refer to our Figure Preparation Guidelines:

8) Please note that supplementary information at EMBO Press has been superseded by the 'Expanded View' for inclusion of additional figures, tables, movies or datasets; with up to five EV Figures being typeset and directly accessible in the HTML version of the article. For details and guidance, please refer to:

embopress.org/page/journal/14602075/authorguide#expandedview

9) To facilitate reproducibility and cross-laboratory adoption of methodologies, please structure the Materials & Methods section as outlined in our guide to authors, including a completed Reagents and Tools Table that can be downloaded from our author guidelines as well (<https://www.embopress.org/page/journal/14602075/authorguide#structuredmethods>).

10) Digital image enhancement is acceptable practice, as long as it accurately represents the original data and conforms to community standards. If a figure has been subjected to significant electronic manipulation, this must be clearly noted in the figure legend and/or the 'Materials and Methods' section. The editors reserve the right to request original versions of figures and the original images that were used to assemble the figure. Finally, we generally encourage uploading of numerical as well as gel/blot image source data; for details see: embopress.org/page/journal/14602075/authorguide#sourcedata

At EMBO Press, we ask authors to provide source data for the main manuscript figures. Our source data coordinator will contact you to discuss which figure panels we would need source data for and will also provide you with helpful tips on how to upload and organize the files.

Further information is available in our Guide For Authors:

In the interest of ensuring the conceptual advance provided by the work, we recommend submitting a revision within 3 months (10th Jun 2025). Please discuss the revision progress ahead of this time with the editor if you require more time to complete the revisions. Use the link below to submit your revision:

Link Not Available

Referee #1:

The Asgard archaea, discovered a little over 10 years ago as the closest living archaeal relatives of eukaryotes, have proven incredibly challenging to isolate and culture, limiting investigations of their cell biology to biochemical reconstitution and experiments in trans. Nonetheless these studies have already yielded fascinating insights into Asgard ESCRT proteins, actins and a much rarer set of tubulins. Building on this base, Kumari et al. carry out a very nice study here on the dual FtsZ genes of Odinararchaeota. The paper is well written, the experiments are well controlled with clearly interpretable results. The divergent filaments in Figure 2 are exceptionally striking!

My main comments have to do with phylogenetic context, interpretation and the discussion. I think all could be addressed with a careful rewriting of sections of the manuscript.

Figure S1 has a FtsZ tree, but this is limited to Asgard archaea and outgroups. What would be much more useful would be to have a larger FtsZ tree which places the two Odin paralogs within a greater prokaryotic cytoskeletal context. This would help with setting the context in the introduction and also with the interpretation of certain key results of the paper. The authors

mention (I imagine convergent) evolution of a FtsZ N-terminal amphipathic helix (for direct membrane insertion) in *Mycoplasma* and also in chloroplasts; in a different section they discuss the possibility of analogies to *Haloferax* division. It would be better to tie these speculations to explicit evolutionary models for how FtsZ sequences have diverged in various prokaryotic lineages in the presence and absence of additional paralogs (which presumably relax some evolutionary constraints on sequence and structure).

The section of the discussion focussed on eukaryogenesis is problematic. The position of the eukaryotic radiation within Asgard archaea is unstable, changing with each new batch of Asgard genomes. In many recent analyses the 'Hod' clade were the closest Asgard lineage, with Odin a relatively distant cousin. Therefore the statement on line 302 is simply incorrect. Even if Odin were to prove to be the closest relative, the time since divergence (2 billion years) combined with the extensive HGT within archaeal and bacterial genes means that absolutely nothing about eukaryogenesis can be inferred from a single set of Asgard proteins. Instead, the authors should focus their attention on what could be learned about ARCHAEAL cell biology - both in Asgard archaea, once the relevant cultured models become available, and other clades of poorly understood archaea.

Related, continuing the theme of problematic/contradictory evolutionary conclusions - on line 298 the authors first point out that FtsZ-SepF interactions are widespread across microbial life - and then later say that this demonstrates "an early tendency towards functional specialisation" and "the early role of accessory proteins".

With just one example of a genome containing 2 FtsZ proteins, the full ESCRT system, and a tubulin gene - Odin - it is not possible to draw conclusions about what this means in terms of cytoskeletal function and division of labour. Especially without the actual cells! Furthermore, there is a whole set of organelle FtsZs that are continuously evolving in the presence of a fully specialised eukaryotic cytoskeleton - what of this? How different/specialised are they?

Referee #2:

In this manuscript, Kumari et al. study the two FtsZ homologs in *Odinarchaeum yellowstonii* LCB_4, as a model to understand FtsZ structure, interactions and dynamics in Asgard archaea, the closest archaeal relatives of eukaryotes. This aim fits well in the current field of Asgard archaeal biology as a novel approach to study eukaryogenesis. The origin of the eukaryotic cytoskeleton remains unclear, and the transition from the archaeal cytoskeleton to the dominance of tubulin and microtubules in eukaryotes remains a key gap in evolutionary cell biology. This study sets some basic understanding for the role of *Odinarchaeal* FtsZs, namely: (1) they describe the domain structure of the two paralogs in *Odinarchaeum*, (2) test their co-interaction and polymerization behavior, and (3) find SepF tethering in FtsZ2 and (4) membrane anchoring in FtsZ1. While still not a comprehensive model of the role of FtsZ in *Odinarchaeum*, this manuscript includes important analyses and will help set a firm basis to understanding the Asgard archaeal cytoskeleton. In my review, I focus predominantly on the evolutionary aspects of this study, for which I only have a few comments.

My main concern of the current version of this manuscript is the superficial contextualisation of the Asgard archaeal FtsZs. The writing in this text almost seems to suggest that the duplication that originated two FtsZ paralogs is unique to Asgard archaea (as may be interpreted from the sequence sampling in phylogenetic trees) and some of the discussion, and sometimes draw very few insights from comparing *Odinarchaeal* FtsZs to other archaeal homologs.

First, the phylogenetic analyses should make it clear that two FtsZ paralogs exist in many archaea (see e.g. Ithurbide et al 2022 *Trends in Microbiol*, Santana-Molina et al 2022 *Front Microbiol*). The analysis performed by the authors suggests that the two paralogs identified in *Odinarchaeum* correspond to the usual archaeal FtsZ1 and FtsZ2, but in order to discard a possible Asgard archaeal-specific duplication event and a different level of homology, the authors should perform a phylogenetic analysis in which they incorporate FtsZ1 and FtsZ2 proteins from across the known archaeal diversity (at least from Euryarchaea and TACK archaea, including Korarchaea, the only TACK archaea with both paralogs in Ithurbide et al 2022).

Moreover, the authors should also indicate the version of IQTree that was used, and how exactly they ran model testing. There are clear incongruences between the FtsZ tree and the established Asgard archaeal species trees. It is notoriously difficult to perform robust and accurate trees of relatively short proteins, but the authors could try better methodology. To start with, the models used by the authors (LG+G4) is very simplistic. I recommend they try complex models (e.g. LG+G4+C60, also available in IQTree), with the hope they may obtain more robust topologies.

Second, all results on domain composition, interactions and function of FtsZ1 and FtsZ2 needs to be explicitly compared with known results from other archaea. For example, how does the SepF anchoring of FtsZ1 identified in Pende et al (2021 *Nat Comms*) fit with the SepF anchoring of FtsZ2 identified here? Are N-terminal amphipathic helices found in FtsZ1 (or FtsZ2) in other archaea? A deeper discussion in this regard seems necessary to identify insights into the evolutionary history of these proteins, a core goal of this manuscript.

Importantly, all data used for phylogenetic analyses (raw sequences, alignments and trees) need to be provided alongside this manuscript.

Minor comments:

Taxonomy (throughout the text): Archaeal and bacterial taxonomy is in a state of flux, with multiple terms used as synonyms for different groups of microbes. I do not mean to push any specific nomenclature, but I would like to point out that many (most?) of the researchers in this field follow GTDB's convention that Asgard archaea are a phylum rather than a superphylum ("Asgardarchaeota") and Odinararchaea are a class rather than a phylum (Odinarchaeia). I am mentioning this in case the authors would wish to clarify, or perhaps even replace the terms in the text. Similarly, the authors often use just the term "Odin" to refer to this lineage, or simply "Asgard" to refer to Asgard archaea, but this terminology should be explicitly introduced as an abbreviation and not a formal name. Additionally, "Candidatus" is sometimes used before "Odinarchaeota" but not always - again, this should be made clearer and consistent.

L16 and L29. The terms "deep-branching" and "transitional" (used again in L303) seem strange in this text. The first one does not convey a clear meaning, and the second is incorrect (Odinarchaea are not transitional themselves). I suggest that the authors refer to the close relatedness between Asgard archaea and eukaryotes in a clearer manner, instead of using these terms. Another error in the abstract is the reason indicated as why Asgard archaea have become models to study eukaryogenesis: it is not only that they encode homologs of typically eukaryotic protein families, but indeed because of their close relatedness with eukaryotes (e.g. the same homologs in more distant relatives would be less interesting to study).

L32-39. Given how fast-evolving this field is, and how many studies have contributed to our understanding of the evolutionary history of Asgard archaea and the presence of eukaryotic signature proteins, I wonder if it may be a good idea to supplement the references in this paragraph with some recent reviews. This is not a request, just a thought.

L62. Archaea are not a superphylum, but a domain.

L66. Here "Asgard archaeon" is used as singular, but Odinarchaeota is used as a phylum including many different organisms, so the phrasing is strange. I suggest simply saying "Odinarchaeota", or somehow rephrasing (e.g. Odinarchaeota are members of the Asgard archaea, and possess (...)).

L74-76. I think the authors convincingly express that studying Odinararchaeal cytoskeletal proteins is an interesting and important goal. However, I find the justification to study only and specifically FtsZ (to the exclusion of CetZ or tubulin) less convincing. I suggest a stronger or clearer emphasis to explain why this study is important (which I do believe is the case).

L99. "kingdoms" is incorrect or unclear here. Perhaps "domains" would be best.

L117. "ancestral relative of tubulin" is erroneous here. The evolutionary link between tubulins and FtsZ does not seem to be one where FtsZ gave rise to tubulins. Instead, the monophyletic group of archaeal FtsZs and the monophyletic group of tubulins are related to each other, in still uncertain ways (but this monophyly does seem supported in multiple evolutionary studies). This incorrect interpretation appears in multiple instances in the text, and should be rectified.

In Figure 1, it is surprising that the authors compare the inferred structures of Odinararchaeal FtsZs with the E. coli structure - while their similarity is very informative, it is a much more distant homolog compared to other archaeal FtsZs. I suggest the authors incorporate in this figure archaeal homologs. Please also clarify the identity of the four proteins displayed in Figure 1B in the figure legend or even in the figure itself.

L264. FtsZ have not transitioned into tubulin-like proteins (as explained above based on phylogenetic arguments). And they have certainly not transitioned into actin-like proteins, as there seems to be no homology between these at all. This is reiterated in L308-310, and should also be rephrased.

L268-269. I do not believe it is correct to claim that the co-existence of tubulin and FtsZ homologs in Odinararchaea is "evidence" of any "shift". This would need to come from functional insights, and not just simultaneous presence.

L275-276. I am not sure the author's analyses "reveal" two distinct FtsZ proteins in Odinararchaea - these were previously revealed, see e.g. Ithurbe et al 2022.

L300. This sentence is misleading - it is not just that assigning FtsZ "as the direct ancestors of the evolved cytoskeletal elements" is "not possible", it would simply be a conceptually flawed statement. Odinararchaeal proteins exist now, and are therefore not the ancestors of any other protein from which they diverged billions of years ago.

L310-312. Is this interaction between FtsZ and other proteins not a common feature of FtsZ across the known prokaryotic diversity? How is it different from the Methanobrevibacter interaction shown by Pende et al (2021 Nat Comms)?

L321-322. While appealing, this narrative that FtsZ cytoskeleton is simple and eukaryotic microtubules are complex does not

seem clear. What is the basis for this difference in complexity?

Suppl. Fig S1 and S3. Accession numbers or strain names need to be shown in these figures, so that the reader can really trace the phylogeny to actual sequences and taxonomy.

Referee #3:

In this study the authors examined the FtsZ proteins in Candidatus Odinararchaeota, belonging to Asgard archaea to understand the evolution and assembly of the FtsZ-Tubulin protein superfamily. Comparative analysis of two FtsZ isoforms, namely OdinFtsZ1 and OdinFtsZ2, shows distinct filament structures and membrane targeting mechanisms. While OdinFtsZ1 assembles into curved single filaments and directly binds to liposomes with its N-terminal amphipathic helix, OdinFtsZ2 polymerises into stacked spiral ring-like structures and is anchored to liposomes with the help of OdinSepF via its C-terminal tail that interacts with OdinSepF. The findings highlight the diversity of FtsZ proteins in Odinararchaeota and more general in archaea, offering new insights into the evolutionary development of tubulin family proteins.

General comment:

I want to congratulate the authors on this well designed and executed study - a nice biochemistry paper that shows protein functioning and structural conservation. The manuscript is well written, just a few things need to be specified or corrected (see comments below). It will be of general interest for Archaeal biologists, structural biologists, scientists working on the tubular super family and evolutionary biologists. Its novelty relies not in the type of experiments used but in the findings of the OdinFtsZ1 and OdinFtsZ2 polymerization pattern. I really enjoyed reading the article and would recommend it for publication!

Specific comments/minor concerns:

- Line 32: One could mention the plum names that were proposed - Asgardarchaeota and Prometeonarchaeota. Asgard archaea is more of a common name.
- Line 36 and 37: Please make sure to write the work homolog/homologue the same way throughout the manuscript
- Line 61: Archaea per se are not a superphylum; please rephrase.
- Figure 1C/D: it is not clear to me why you used two different concentrations of FtsZ1 (4 μ M) and FtsZ2 (10 μ M). In order to compare the two plot the amount used should be the same. Also the plot scale should be the same to have a good comparison. Here it is 2 AU vs 20AU. To me it could be that the 10 μ M + Mg control has a similar pattern like the 4 μ M + Mg control, but it is not visible due to the scale difference. I would say that in FtsZ1 very little polymerization/ light scattering could be observed. To really draw a conclusion you would have to show the two FtsZs in both conditions (4 and 10 μ M) and then compare them. Which I would recommend the authors to do.
- Line 165: The fact that FtsZ1 formed short filaments even without nucleotides, could explain partly the strange polymerization pattern Fig 1C. The authors should prepare the buffer, add the protein and immediately monitor the 90° light scattering to see if there is a slight shift in the base line with time.
- Figure 2 A: the contrast could be increased here to resemble more the image in C and for better visibility of the filaments.
- Line 207 and 208: again archaea are not a superphylum, but a domain.
- Figure 3: How come that in this figure the SepF runs as a lower band when compared to Supp. Fig 1C? Same is true for the FtsZs. Could it be that the marker is wrongly annotated in one or the other figure?
- Figure 4B: I don't think this panel contributes much to the figure. I would be better showing that the N-ter binds the liposomes vs the truncated version that doesn't bind. This would be a more plakative representation of the results than just the two FtsZs with the liposome.
- Line 326- 327: "It is possible that like in the archaeon Haloferax13, the two Odin FtsZs with distinct properties might cooperate in cell division" It would be nice to test if OdinFtsZ could complement Haloferax FtsZ KO. How similar are the sequences from *H. volcanii* compared to the Odin FtsZ and also SepF? Could the *H. volcanii* SepF interact with the OdinFtsZ2 and vice versa? Without a in vivo experiment it is hard to really attribute the function in cell division of the OdinFtsZ . However, in vivo experiments with *H. volcanii* might be out of scope for this study, but would be nice to mention in the discussion.
- Line 368 -369: it seems like there are some things missing in the lysis buffer. It should contain at least lysozyme. Or the cells should have been lysed by sonication. Otherwise I cannot imagine how the cells would have been lysed with a simple buffer just containing 50 mM Tris pH 8, 200 mM NaCl, 10% glycerol.
- Line 379-386: Why was the protein flash-frozen and stored at -80°C before the ion exchange? This seems strange.
- Line 430-439: Was the quality of the liposome preparation somehow checked? Maybe with negative stain EM to see if the liposomes really formed.
- Line 478: Why was the concentration of OdinFtsZ1 and FtsZ2 different? Here the authors used more for FtsZ1 than FtsZ2, whereas in the 90° light scattering assay they used less for FtsZ1 than FtsZ2. This is very counter intuitive and might introduce some concentration dependent patterns.
- Line 571: It would be good to have a list with the sequences of the used primers as well.

- All figures with gels: Please put in the figure legend what is the expected size of the different purified proteins. It is nowhere mentioned and should be mentioned at least once when you describe the proteins for the first time.

Referee #4:

In this work, Kumari, Uthaman et al. identify two FtsZ isoforms co-occurring together with tubulin in *Odinarchaeota*. The authors characterize differences in the polymerization and membrane tethering properties of the two FtsZ isoforms and investigate their divergent protofilament structures by cryoEM. While I am no expert in the field of eukaryotic cellular evolution, the comparison of the two FtsZ isoforms seems to provide insights into the evolution and diversification of the eukaryotic cytoskeleton.

I am serving as a technical reviewer for the cryoEM aspects of the manuscript. Generally, cryoEM imaging follows state-of-the-art procedures and cryoEM data seem of high quality as far as it can be judged from the data presented. However, analysis of the cryoEM data is very superficial, should be better described and most likely falls far behind what could be achieved with the available data.

There are several points that should be addressed in a revised version of the manuscript:

- 1) The authors should provide much more detail on cryoEM image analysis in the methods section. Also, it is good practice to provide a processing scheme in which the number of micrographs, the number of particles, the individual processing steps, software used etc. is being summarized. Please provide such a scheme as a supplementary figure.
- 2) The authors should try to process their cryoEM data beyond the level of 2D classification and move on to 3D reconstruction, at least for FtsZ1. Their 2D classes indicate that at least a medium resolution 3D reconstruction could be achievable, which would be highly informative and substantially strengthen the manuscript.
- 3) Line 178: "Unlike the 2D class averages of *OdinFtsZ1*, which show high-resolution features, the 2D classes of *OdinFtsZ2* show low-resolution features of the ring and no clear indication of subunit stoichiometry indicating heterogeneity." How do the authors come to this conclusion? At least based on what is shown in Fig. 2, the 2D classification for FtsZ1 and FtsZ2 has been performed at strongly differing pixel sizes and box sizes. FtsZ1 boxes contain only a few asymmetric units, while FtsZ2 particles are 'zoomed out' much more and contain entire rings of dozens of particles. One would likely not expect 2D classes of the quality as for FtsZ1 at this level. The authors should repeat 2D classification for FtsZ2 using pixel sizes and box sizes as for FtsZ1.
- 4) Lines 163-165: Could the authors quantify the length of FtsZ1 filaments? In any case, it should become clear from the phrasing that the filaments in absence of any nucleotide are shorter as compared to the other conditions, which is not the case right now.
- 5) Line 115: "Though *OdinFtsZ2* clusters along with other FtsZ2 members (Fig. S1A), it possesses an insertion of a loop in the CTD, which is not found in other FtsZ proteins." Please support this claim by a multiple sequence alignment.
- 6) Lines 189-192: The authors point out a clear discrepancy between the pelleting behavior of FtsZ2 and its oligomeric state observed by EM. The authors should try to resolve this discrepancy or discuss in the text what could be possible reasons.
- 7) The same centrifugal force and pelleting times are used in the filament pelleting and liposome pelleting assays. How can the authors differentiate pelleting due to liposome association from filament assembly in the liposome pelleting assay? In particular FtsZ1 was shown in this study to oligomerize independent from nucleotides. Please explain/discuss in the text.

Minor points:

- 1) Figs 2A-D need scale bars.
- 2) For Fig S1, figure callouts in the text, panel labeling and figure captions/legends seem to be mixed up. Please correct.
- 3) Fig. 1A: please label the different FtsZ segments in the panel, instead of just color-coding them.
- 4) Fig. 2A,C: can the micrographs in Fig. 2 be shown at a similar contrast?
- 5) Fig. 3: please label the identity of bands in SDS-PAGE gels.
- 6) Fig. 4A: red box not required?
- 7) Lines 49-51: It is not quite clear what the authors refer to here. Do they refer to the growth direction as compared to tubulin? Please rephrase.
- 8) Line 175: "In a given field of view, the number of longer stacked rings was less, but shorter stacks were more prominent." It is unclear to me what the authors relate to here. Do the authors refer to the number of rings per stacked assembly, the diameter of rings, etc? Please rephrase.

EMBOJ2025-120418

Kumari et al., Distinct filament morphology and membrane tethering features of the dual FtsZs in Odinararchaeota

Point by Point Response to Referees

Referee #1:

The Asgard archaea, discovered a little over 10 years ago as the closest living archaeal relatives of eukaryotes, have proven incredibly challenging to isolate and culture, limiting investigations of their cell biology to biochemical reconstitution and experiments in trans. Nonetheless these studies have already yielded fascinating insights into Asgard ESCRT proteins, actins and a much rarer set of tubulins. Building on this base, Kumari et al. carry out a very nice study here on the dual FtsZ genes of Odinararchaeota. The paper is well written, the experiments are well controlled with clearly interpretable results. The divergent filaments in Figure2 are exceptionally striking!

My main comments have to do with phylogenetic context, interpretation and the discussion. I think all could be addressed with a careful rewriting of sections of the manuscript.

- Figure S1 has a FtsZ tree, but this is limited to Asgard archaea and outgroups. What would be much more useful would be to have a larger FtsZ tree which places the two Odin paralogs within a greater prokaryotic cytoskeletal context. This would help with setting the context in the introduction and also with the interpretation of certain key results of the paper. The authors mention (I imagine convergent) evolution of a FtsZ N-terminal amphipathic helix (for direct membrane insertion) in *Mycoplasma* and also in chloroplasts; in a different section they discuss the possibility of analogies to *Haloferax volcanii* cell division. It would be better to tie these speculations to explicit evolutionary models for how FtsZ sequences have diverged in various prokaryotic lineages in the presence and absence of additional paralogs (which presumably relax some evolutionary constraints on sequence and structure).

Response: We appreciate the reviewer for pointing out the limited taxonomic scope of FtsZ phylogeny in Figure S1. We have now included an expanded phylogenetic analysis which has broader representation of prokaryotic FtsZ homologs across different phyla of bacteria and archaea along with sequences from cyanobacteria (**Fig. EV1**). This has been discussed in the result section [Page: 6, line no: 131- 136].

While the evolutionary basis of having two distinct membrane binding mechanism remains unclear, We have revised the text to more explicitly discuss the direct membrane-binding ability of OdinFtsZ1, drawing comparison with FtsZs from *Mycoplasma* and chloroplast as well as the anchor mediated membrane tethering mechanism observed in *H. volcanii* [Page: 14-15, line no: 368-398].

- The section of the discussion focussed on eukaryogenesis is problematic. The position of the eukaryotic radiation within Asgard archaea is unstable, changing with each new batch of Asgard genomes. In many recent analyses the 'Hod' clade were the closest Asgard lineage, with Odin a relatively distant cousin. Therefore the statement on line 302 is simply incorrect. Even if Odin were to prove to be the closest relative, the time since divergence (2 billion years) combined with the extensive HGT within archaeal and bacterial genes means that absolutely nothing about eukaryogenesis can be inferred from a single set of Asgard proteins. Instead, the authors should focus their attention on what could be learned about ARCHAEAL cell biology - both in Asgard archaea, once the relevant cultured models become available, and other clades of poorly understood archaea.

Response: We thank the reviewer for this suggestion regarding the discussion section and to shift the emphasis from eukaryogenesis to Asgard cell biology. We acknowledge the phylogenetic instability in Asgard and therefore reframed this section of discussion to discuss how the structural feature of OdinFtsZ paralogs contribute to our current understanding of cytoskeleton system in Asgard archaea. Accordingly, we have removed any statements that imply any implications of the findings reported in this manuscript to the evolutionary origin of eukaryotic cytoskeletal complexity. The line 302 "our findings support the hypothesis that the eukaryotic cytoskeleton evolved through functional diversification and innovation within the transitional archaeal phylum Odinararchaeota" has been modified and new discussion section is added [Page: 15-16, line no: 399-434].

We would also like to clarify that our use of the term "transitional" in reference to Odinararchaeota was not intended to imply a direct ancestral link to eukaryotes. Rather, we used the term in a more functional sense, to highlight the unique coexistence of both FtsZ and tubulin homologs within the same archaeal lineage. This combination is of particular interest given that similar dual systems are now also supported by recent findings in Lokiarchaeota, where two distinct tubulin protofilament assemblies have also been described (Wollweber *et al*, 2025). This text and the phrasing is now removed from the manuscript.

- Related, continuing the theme of problematic/contradictory evolutionary conclusions - on line 298 the authors first point out that FtsZ-SepF interactions are widespread across microbial life - and then later say that this demonstrates "an early tendency towards functional specialisation" and "the early role of accessory proteins"

Response: We acknowledge this inconsistency, and the section has been removed. Instead the paragraph now discusses this interaction in comparison to the previously reported interactions between FtsZ and membrane anchor proteins [Page: 14, line no: 374-390].

- With just one example of a genome containing 2 FtsZ proteins, the full ESCRT system, and a tubulin gene - Odin - it is not possible to draw conclusions about what this means in terms of cytoskeletal function and division of labour. Especially without the actual cells! Furthermore, there is a whole set of organelle

FtsZs that are continuously evolving in the presence of a fully specialised eukaryotic cytoskeleton - what of this? How different/specialised are they ?

Response: We agree with the reviewers that firm conclusions about the cytoskeleton function and division of labour cannot be drawn from a single genome, especially in the absence of cellular data. Accordingly, we have revised any sections of the discussion that previously stated this as a definitive conclusion.

Further, we have now included organelle FtsZs also in the phylogenetic tree given in **Fig. EV1** of the revised manuscript. Additionally, we have discussed organelle-localized FtsZ proteins in eukaryotes, which have continued to evolve alongside eukaryotic cytoskeleton and co-operate in organelle division. These systems provide valuable comparative examples of how FtsZ can persist and diversify functionally in the presence of other cytoskeletal elements. [Page: 13, line no: 335-338].

Referee #2:

In this manuscript, Kumari et al. study the two FtsZ homologs in *Odinarchaeum yellowstonii* LCB_4, as a model to understand FtsZ structure, interactions and dynamics in Asgard archaea, the closest archaeal relatives of eukaryotes. This aim fits well in the current field of Asgard archaeal biology as a novel approach to study eukaryogenesis. The origin of the eukaryotic cytoskeleton remains unclear, and the transition from the archaeal cytoskeleton to the dominance of tubulin and microtubules in eukaryotes remains a key gap in evolutionary cell biology. This study sets some basic understanding for the role of *Odinarchaeal* FtsZs, namely: (1) they describe the domain structure of the two paralogs in *Odinarchaeum*, (2) test their co-interaction and polymerization behavior, and (3) find SepF tethering in FtsZ2 and (4) membrane anchoring in FtsZ1. While still not a comprehensive model of the role of FtsZ in *Odinarchaeum*, this manuscript includes important analyses and will help set a firm basis to understanding the Asgard archaeal cytoskeleton. In my review, I focus predominantly on the evolutionary aspects of this study, for which I only have a few comments.

My main concern of the current version of this manuscript is the superficial contextualisation of the Asgard archaeal FtsZs. The writing in this text almost seems to suggest that the duplication that originated two FtsZ paralogs is unique to Asgard archaea (as may be interpreted from the sequence sampling in phylogenetic trees) and some of the discussion, and sometimes draw very few insights from comparing *Odinarchaeal* FtsZs to other archaeal homologs.

- First, the phylogenetic analyses should make it clear that two FtsZ paralogs exist in many archaea (see e.g. Ithurbide et al 2022 *Trends in Microbiol*, Santana-Molina et al 2022 *Front Microbiol*). The analysis performed by the authors suggests that the two paralogs identified in *Odinarchaeum* correspond to the usual archaeal FtsZ1 and FtsZ2, but in order to discard a possible Asgard archaeal-specific duplication event and a different level of homology, the authors should perform a phylogenetic analysis in which they incorporate FtsZ1 and FtsZ2 proteins from across the known archaeal diversity (at least from Euryarchaea and TACK archaea, including Korarchaea, the only TACK archaea with both paralogs in Ithurbide et al 2022).

Response: We thank the reviewer for pointing out the importance of placing *Odinarchaeota* FtsZ paralogs within the broader context of archaeal FtsZ proteins. We really appreciate the reviewers reference to the comprehensive analyses by Ithurbide et al. (2022) and Santana-Molina et al. (2022). We have now modified the tree (**Fig. EV1**) with representative sequences from different archaeal phyla along with bacterial FtsZ sequences. [Page: 6, line no: 131-136].

- Moreover, the authors should also indicate the version of IQTree that was used, and how exactly they ran model testing. There are clear incongruences between the FtsZ tree and the established Asgard archaeal species trees. It is notoriously difficult to perform robust and accurate trees of relatively short proteins, but the authors could try better methodology. To start with, the models used by the authors (LG+G4) is very simplistic. I recommend they try complex models (e.g.

LG+G4+C60, also available in IQTree), with the hope they may obtain more robust topologies.

Response: We have updated the methods section for phylogenetic analysis to include the details of IQTREE version and the model testing. For the SepF phylogeny, we re-evaluated the model and found that LG+G4+C60 provided robust and well-supported results. We have accordingly reconstructed the SepF tree using this model.

For the FtsZ phylogeny, we tested the LG+G4+C60 model as recommended. However, model testing based on BIC supported the LG+G4+I model over more complex alternatives. Furthermore, the LG+G4+C60 model yielded topologies with lower bootstrap support and an incongruent placement of Korarchaeota (see Response Fig. 2). In contrast, the LG+G4+I model resulted in a more stable and interpretable tree (Response Fig. 1). Given the known challenges in reconstructing reliable phylogenies from short and conserved proteins like FtsZ, we opted to retain the LG+G4+I model, balancing both model fit and topological robustness. We have included both trees in our response for transparency and have highlighted the key discrepancies.

LG+G4+I

Response Fig. 1. FtsZ tree constructed using LG+G4+I model

Response Fig. 2. FtsZ tree constructed using LG+G4+ C60 model

- Second, all results on domain composition, interactions and function of FtsZ1 and FtsZ2 needs to be explicitly compared with known results from other archaea. For example, how does the SepF anchoring of FtsZ1 identified in Pende et al (2021 Nat Comms) fit with the SepF anchoring of FtsZ2 identified here? Are N-terminal amphipathic helices found in FtsZ1 (or FtsZ2) in other archaea? A deeper discussion in this regard seems necessary to identify insights into the evolutionary history of these proteins, a core goal of this manuscript.

Response: We thank the reviewer for this important suggestion. We agree that a deeper comparison of our findings with known results from other archaeal systems is essential to contextualize the domain composition, interactions, and potential functional specializations of Odinararchaeota FtsZ1 and FtsZ2.

In response, we have expanded the discussion to include explicit comparisons with previously characterized archaeal FtsZ systems. In particular, we now discuss the findings in *M.smithii* (Pende *et al*, 2021), which demonstrated SepF as an anchor for FtsZ. We have also compared this with the known FtsZ2-SepF interaction in *H. volcanii* (Nußbaum *et al*, 2021) [Page: 14-15, line no: 368-408].

We have also examined the presence of N-terminal amphipathic helices in FtsZ paralogs across archaea based on the sequence alignment of archaeal FtsZ sequences (**Fig. S1A**) (Liao, *et al*, 2021, *Nat. Microbiol*) and included a relevant comment in the discussion section of the manuscript [Page: 14, line no: 368-374].

- Importantly, all data used for phylogenetic analyses (raw sequences, alignments and trees) need to be provided alongside this manuscript.

Response: We appreciate the reviewer's emphasis on data accessibility. All the raw files (with sequence accession no), aligned fasta files and trees have been uploaded as part of the source file for **Fig. EV1** and **Fig. EV5**.

Minor comments:

- Taxonomy (throughout the text): Archaeal and bacterial taxonomy is in a state of flux, with multiple terms used as synonyms for different groups of microbes. I do not mean to push any specific nomenclature, but I would like to point out that many (most?) of the researchers in this field follow GTDB's convention that Asgard archaea are a phylum rather than a superphylum ("Asgardarchaeota") and Odinararchaea are a class rather than a phylum (Odinararchaeia). I am mentioning this in case the authors would wish to clarify, or perhaps even replace the terms in the text. Similarly, the authors often use just the term "Odin" to refer to this lineage, or simply "Asgard" to refer to Asgard archaea, but this terminology should be explicitly introduced as an abbreviation and not a formal name. Additionally, "Candidatus" is sometimes used before "Odinararchaeota" but not always- again, this should be made clearer and consistent.

Response: We thank the reviewer for pointing this out. According to the Genome Taxonomy Database (GTDB), Asgardarchaeota—commonly referred to as Asgard archaea—is a candidate phylum-level archaeal clade. We have followed

this taxonomic nomenclature system, and it is now followed throughout the text. Additionally, We have ensured consistency for use of the "Odinarchaeota" throughout the revised manuscript.

- L16 and L29. The terms "deep-branching" and "transitional" (used again in L303) seem strange in this text. The first one does not convey a clear meaning, and the second is incorrect (Odinarchaea are not transitional themselves). I suggest that the authors refer to the close relatedness between Asgard archaea and eukaryotes in a clearer manner, instead of using these terms. Another error in the abstract is the reason indicated as why Asgard archaea have become models to study eukaryogenesis: it is not only that they encode homologs of typically eukaryotic protein families, but indeed because of their close relatedness with eukaryotes (e.g. the same homologs in more distant relatives would be less interesting to study).

Response: We thank the reviewer for the comment. We have now revised the wording in both the abstract and the main text to emphasize the relatedness of eukaryotes and Asgard archaea, avoiding terms such as 'transitional' and 'deep-branching'.

- L32-39. Given how fast-evolving this field is, and how many studies have contributed to our understanding of the evolutionary history of Asgard archaea and the presence of eukaryotic signature proteins, I wonder if it may be a good idea to supplement the references in this paragraph with some recent reviews. This is not a request, just a thought.

Response: We appreciate the reviewer's helpful recommendation. Indeed, this is a rapidly evolving field, with new studies continually expanding our understanding of Asgard archaea. We agree that including recent reviews offers a broader perspective. We have now supplemented the new references in introduction paragraph (Eme *et al*, 2017; Spang *et al*, 2018; Lu *et al*, 2020; Stairs & Ettema, 2020; Vosseberg *et al*, 2024; Zhang *et al*, 2025) with several recent reviews that synthesize the current understanding of Asgard archaea and their evolutionary implications [Page: 3, line no: 36-52].

- L62. Archaea are not a superphylum, but a domain.

Response: Thank you for pointing this out. In this particular line, our reference to "superphylum" was intended to describe specific archaeal clades such as TACK, DAPANN, not the entire archaeal domain. We have revised the wording in the manuscript to clarify this distinction [Page: 4, line no: 79-82].

- L66. Here "Asgard archaeon" is used as singular, but Odinarchaeota is used as a phylum including many different organisms, so the phrasing is strange. I suggest simply saying "Odinarchaeota", or somehow rephrasing (e.g. Odinarchaeota are members of the Asgard archaea, and possess (...)).

Response: Thanks, this has been corrected in the manuscript [Page: 4, line no: 87-90].

- L74-76. I think the authors convincingly express that studying Odinararchaeal cytoskeletal proteins is an interesting and important goal. However, I find the justification to study only and specifically FtsZ (to the exclusion of CetZ or tubulin) less convincing. I suggest a stronger or clearer emphasis to explain why this study is important (which I do believe is the case).

Response: We thank the reviewer for their encouraging comments and for highlighting the need to more clearly justify the focus on FtsZ. In the revised manuscript, we have now emphasized that while multiple cytoskeletal proteins, including tubulins, are present in Asgard archaea, FtsZ remains the central cell division driver across most prokaryotic systems.

Moreover, although FtsZ homologs have been identified in several Asgard archaeal genomes, there has been no experimental characterization of Asgard FtsZ to date. Our study represents the first effort to investigate the properties of an Asgard FtsZ, offering a critical step toward understanding the functional diversity of division machinery in archaea. We believe this study fills an important gap in the field and lays the groundwork for future comparative studies on other cytoskeletal systems in Asgard. A detailed explanation on the choice of FtsZ as the subject of this study has been described in the revised manuscript [Page: 4, line no: 79-86; [Page: 5, line no: 100-107; Page: 13, line no: 338-343].

- L99. "kingdoms" is incorrect or unclear here. Perhaps "domains" would be best.

Response: Thanks, this has been modified in the revised manuscript [Page: 6, line no: 128].

- L117. "ancestral relative of tubulin" is erroneous here. The evolutionary link between tubulins and FtsZ does not seem to be one where FtsZ gave rise to tubulins. Instead, the monophyletic group of archaeal FtsZs and the monophyletic group of tubulins are related to each other, in still uncertain ways (but this monophyly does seem supported in multiple evolutionary studies). This incorrect interpretation appears in multiple instances in the text, and should be rectified.

Response: We fully agree with the reviewer and have corrected any potential misinterpretation in the revised manuscript.

- In Figure 1, it is surprising that the authors compare the inferred structures of Odinararchaeal FtsZs with the *E. coli* structure - while their similarity is very informative, it is a much more distant homolog compared to other archaeal FtsZs. I suggest the authors incorporate in this figure archaeal homologs. Please also clarify the identity of the four proteins displayed in Figure 1B in the figure legend or even in the figure itself.

Response: We thank the reviewer for this suggestion. We agree that including structural comparisons with archaeal homologs provides a more meaningful evolutionary context. In response, we have revised **Fig. 1A** to include the archaeal FtsZ from *Methanococcus jannaschii* (MjFtsZ1; PDB ID: 1FSZ) alongside *Escherichia coli* FtsZ (EcFtsZ; PDB ID: 6UMK) for comparison with Odinararchaeal FtsZs. Additionally, we have included AlphaFold-predicted structures of *Haloferax volcanii* FtsZ1 and FtsZ2, as well as the crystal structure of CetZ, in Appendix supplementary **Fig. S1B**, [Page: 6, line no: 136-140].

We have also clarified the identity of all four proteins displayed in **Fig.1B** by updating the figure legend. We believe these changes enhance the interpretability and relevance of the figure.

- L264. FtsZ have not transitioned into tubulin-like proteins (as explained above based on phylogenetic arguments). And they have certainly not transitioned into actin-like proteins, as there seems to be no homology between these at all. This is reiterated in L308-310, and should also be rephrased.

Response: We appreciate this clarification and have carefully revised the manuscript to avoid any implication that FtsZ proteins have directly transitioned into tubulin or actin.

- L268-269. I do not believe it is correct to claim that the co-existence of tubulin and FtsZ homologs in Odinararchaea is "evidence" of any "shift". This would need to come from functional insights, and not just simultaneous presence.

Response: We thank the reviewer for this important clarification. We agree that the simultaneous presence of tubulin and FtsZ homologs in Odinararchaea does not, on its own, constitute evidence of an evolutionary shift. Our intent was to highlight the diversity of tubulin-like proteins observed across Asgard genomes rather than to imply a definitive evolutionary transition. We have carefully revised the text to avoid overinterpretation and to ensure our statements reflect the observations without overstating their evolutionary implications.

- L275-276. I am not sure the author's analyses "reveal" two distinct FtsZ proteins in Odinararchaea - these were previously revealed, see e.g. Ithurbe et al 2022.

Response: We thank the reviewer for pointing this out. We acknowledge that the presence of two FtsZ homologs in Odinararchaea has been previously reported (Zaremba-Niedzwiedzka *et al*, 2017; Tamarit *et al*, 2022; Ithurbe *et al*, 2022). In our study, we do not claim to have identified them for the first time, but rather focus on characterizing their structural and biochemical differences. We have modified the text accordingly to better reflect this in our revised manuscript [Page: 4, line no: 87-90; Page: 13, line no: 344-345].

- L300. This sentence is misleading - it is not just that assigning FtsZ "as the direct ancestors of the evolved cytoskeletal elements" is "not possible", it would simply be a conceptually flawed statement. exist now, and are therefore not the ancestors of any other protein from which they diverged billions of years ago.

Response: We fully agree with the reviewer that extant Odinararchaeal proteins cannot be considered direct ancestors of eukaryotic cytoskeletal proteins. In response, we have revised the relevant paragraph to remove all references to ancestral or transitional status. Instead, we now frame OdinFtsZ1 and OdinFtsZ2 as functionally distinct homologs that expand the known diversity of FtsZ behaviours and contribute to our understanding of cytoskeletal organisation strategies.

- L310-312. Is this interaction between FtsZ and other proteins not a common feature of FtsZ across the known prokaryotic diversity? How is it different from the *Methanobrevibacter* interaction shown by Pende et al (2021 Nat Comms)?

Response: We agree with the reviewer that interactions between FtsZ and accessory proteins are widespread, and our study serves as a comparative example that contributes to the understanding of known membrane recruitment mechanisms in prokaryotes. Accordingly, we have removed the previous discussion section and rephrased our findings in the context of existing literature.

For example, as described in (Pende *et al*, 2021), *Methanobrevibacter smithii* FtsZ interacts with the membrane anchor protein SepF. However, this organism carries only a single isoform of FtsZ that interacts with SepF. In contrast, in organisms like *Haloferax volcanii*, SepF interacts specifically with FtsZ2, and the mechanism of FtsZ1 recruitment to the membrane remained unknown. We have identified a novel mechanism of membrane recruitment via an N-terminal amphipathic helix in FtsZ1. Similar mechanisms need to be explored in other systems that encode two FtsZ isoforms [Page: 14, line no: 368-390].

- L321-322. While appealing, this narrative that FtsZ cytoskeleton is simple and eukaryotic microtubules are complex does not seem clear. What is the basis for this difference in complexity?

Response: We appreciate this important point. In response, we have removed any language implying a linear or hierarchical view of cytoskeletal complexity. The revised paragraph avoids characterising the Odinarcheal FtsZs concerning eukaryotic cytoskeletal proteins and instead focuses on their distinct biochemical properties and phylogenetic placement among prokaryotic homologs.

- Suppl. Fig S1 and S3. Accession numbers or strain names need to be shown in these figures, so that the reader can really trace the phylogeny to actual sequences and taxonomy.

Response: We have updated the phylogenetic trees to include accession numbers for all sequences used in the analysis. These updated figures are presented in **Fig. EV1** and **Fig. EV5**. The accession numbers, raw sequences, aligned fasta files are also provided in the source data files for these figures.

Referee #3:

In this study the authors examined the FtsZ proteins in Candidatus Odinararchaeota, belonging to Asgard archaea to understand the evolution and assembly of the FtsZ-Tubulin protein superfamily. Comparative analysis of two FtsZ isoforms, namely OdinFtsZ1 and OndinFtsZ2, shows distinct filament structures and membrane targeting mechanisms. While OdinFtsZ1 assembles into curved single filaments and directly binds to liposomes with its N-terminal amphipathic helix, OdinFtsZ2 polymerises into stacked spiral ring-like structures and is anchored to liposomes with the help of OdinSepF via its C-terminal tail that interacts with OdinSepF. The findings highlight the diversity of FtsZ proteins in Odinararchaeota and more general in archaea, offering new insights into the evolutionary development of tubulin family proteins.

General comment:

I want to congratulate the authors on this well designed and executed study - a nice biochemistry paper that shows protein functioning and structural conservation. The manuscript is well written, just a few things need to be specified or corrected (see comments below). It will be of general interest for archaeal biologists, structural biologists, scientists working on the tubular super family and evolutionary biologists. Its novelty relies not in the type of experiments used but in the findings of the OdinFtsZ1 and OdinFtsZ2 polymerization pattern. I really enjoyed reading the article and would recommend it for publication!

Specific comments/minor concerns:

- Line 32: One could mention the plum names that were proposed - Asgardarchaeota and Prometeonarchaeota. Asgard archaea is more of a common name.

Response: We thank the reviewer for the suggestion. In the revised manuscript, we have followed the GTDB nomenclature and introduced Asgardarchaeota phylum at first mention. However, for the sake of readability and consistency with common usage in the literature, we used "Asgard archaea" as a general term to refer to this group throughout the text.

- Line 36 and 37: Please make sure to write the work homolog/homologue the same way throughout the manuscript.

Response: Thanks, this has been modified.

- Line 61: Archaea per se are not a superphylum; please rephrase.

Response: Thanks, this has been modified.

- Figure 1C/D: it is not clear to me why you used two different concentrations of FtsZ1 (4 μ M) and FtsZ2 (10 μ M). In order to compare the two plots the amount used should be the same. Also the plot scale should be the same to have a good comparison. Here it is 2 AU vs 20AU. To me it could be that the 10 μ M + Mg control has a similar pattern like the 4 μ M + Mg control, but it is not visible due to the scale difference. I would say that in FtsZ1 very little polymerization/ light scattering could be observed. To really draw a conclusion you would have to

show the two FtsZs in both conditions (4 and 10 μM) and then compare them. Which I would recommend the authors to do

Response: We thank the reviewer for this helpful observation. We agree that the use of different concentrations and Y-axis scales in Fig. 1C and 1D made it difficult to directly compare the polymerization behavior of OdinFtsZ1 and OdinFtsZ2. Our initial concentrations were chosen based on preliminary optimization to ensure detectable signals for each protein individually; however, we recognize the importance of performing matched-condition comparisons for drawing reliable conclusions.

In response, we have now repeated the light scattering experiments using both OdinFtsZ1 and OdinFtsZ2 at an identical concentration of 5 μM . The revised data, presented in the updated **Fig. 1C**, **Fig. 1D** (replaced the old fig. 1C and 1D), use consistent Y-axis scaling to enable a direct and meaningful comparison. These results reinforce our initial observations: OdinFtsZ2 shows a higher light scattering intensity, while OdinFtsZ1 exhibits lesser intensity under the same conditions. We appreciate the reviewer's suggestion, which has led to a more rigorous and interpretable presentation of this dataset.

We have included the earlier version of the figure as a supplementary figure (**Fig. S2B**, **S2C**) since we felt that the scattering profile at different concentrations convey clear information on the differences in the polymerization dynamics between the two FtsZs [Page: 6-7, line no: 151-164].

- Line 165: The fact that FtsZ1 formed short filaments even without nucleotides, could explain partly the strange polymerization pattern Fig 1C. The authors should prepare the buffer, add the protein and immediately monitor the 90° light scattering to see if there is a slight shift in the base line with time.

Response: We thank the reviewer for the suggestion. However, the initial 400 sec in the light scattering data shows the scatter from the protein alone, before addition of the nucleotide. It can be observed that there is no shift in the baseline unless Mg^{2+} is added.

In addition, the major oligomeric status of the protein is not monomeric when tested by size exclusion chromatography (SEC; Response Fig. 3). The negative stain electron microscopy images (**Fig. S2D**) qualify this data further that FtsZ1

Response Fig. 3. Size exclusion chromatography profile for FtsZ1 tested on a Superdex75 column. Majority of the protein elutes in the void volume, beyond the size range of 70 kDa.

exists as short filaments in solution.

- Figure 2 A: the contrast could be increased here to resemble more the image in C and for better visibility of the filaments.

Response: Thanks, The figure has been improved as suggested.

- Line 207 and 208: again archaea are not a superphylum, but a domain.

Response: Thanks, This has been modified.

- Figure 3: How come that in this figure the SepF runs as a lower band when compared to Supp. Fig 1C? Same is true for the FtsZs. Could it be that the marker is wrongly annotated in one or the other figure?

Response: We thank the reviewer for pointing this out. We have now corrected the marker labels. The updated figure is now moved to Appendix **Fig. S2A** and is included in the revised version of the manuscript. We apologize for the oversight and appreciate the reviewer's attention to detail, which helped us resolve this error. Additionally, for all the purification profile gels (Appendix **Fig. S2A** and Appendix **Fig. S3C**), we have now included the expected sizes of the protein bands as well.

- Figure 4B: I don't think this panel contributes much to the figure. I would be better showing that the N-ter binds the liposomes vs the truncated version that doesn't bind. This would be a more plakative representation of the results than just the two FtsZs with the liposome.

Response: We thank the reviewer for this suggestion and removed the schematic from **Fig. 4B**. In response, we have revised the figure to replace new figure panel **Fig. 4B** with liposome binding of full-length OdinFtsZ1 and **Fig. 4D** with the truncated version lacking the N-terminal region.

- Line 326- 327: "It is possible that like in the archaeon *Haloferax*, the two Odin FtsZs with distinct properties might cooperate in cell division" It would be nice to test if OdinFtsZ could complement *Haloferax* FtsZ KO. How similar are the sequences from *H. volcanii* compared to the Odin FtsZ and also SepF? Could the *H. volcanii* SpeF interact with the OdinFtsZ2 and vice versa? Without a *in vivo* experiment it is hard to really attribute the function in cell division of the OdinFtsZ. However, *in vivo* experiments with *H. volcanii* might be out of scope for this study, but would be nice to mention in the discussion.

Response: We thank the reviewer for this insightful comment and agree that *in vivo* complementation experiments in *Haloferax volcanii* would provide a powerful approach to test whether Odin FtsZs can functionally substitute for native archaeal FtsZ proteins. While such experiments are beyond the current scope of this study, we now highlight in the revised discussion that complementation or co-expression studies in genetically tractable archaeal models like *H. volcanii* could serve as a promising future approach to investigate the functionality and potential division roles of Odin FtsZ paralogs [Page: 15, line no: 399-408].

We also appreciate the suggestion to examine sequence similarity between Odinarchaeaota and *H. volcanii* FtsZs and SepF. Based on the sequence features of *H. volcanii* FtsZs (refer the multiple sequence alignment (**Fig. S1A**), it

is possible that OdinFtsZ1 and OdinFtsZ2 might be able to complement Haloferax FtsZ knockout.

- Line 368 -369: it seems like there are some things missing in the lysis buffer. It should contain at least lysozyme. Or the cells should have been lysed by sonication. Otherwise I cannot imagine how the cells would have been lysed with a simple buffer just containing 50 mM Tris pH 8, 200 mM NaCl, 10% glycerol.

Response: We thank the reviewer for catching this oversight. We have corrected the methods section [Page: 17, line no: 458] to clearly state that cells were lysed by sonication after resuspension in the lysis buffer.

- Line 379-386. Why was the protein flash-frozen and stored at -80°C before the ion exchange? This seems strange.

Response: We thank the reviewer for this question. To clarify, not all protein preparations include ion exchange purification. In cases where only Ni-NTA purification was performed, the proteins were flash-frozen and stored at -80°C . However, for preparations that included ion exchange chromatography, the proteins were processed directly after Ni-NTA purification without freezing. We have now revised the relevant lines in the Methods section to clarify this. We appreciate the reviewer's comment, which helped us clarify this [Page: 18, line no: 467-476].

- Line 430-439: Was the quality of the liposome preparation somehow checked? Maybe with negative stain EM to see if the liposomes really formed.

Response: We thank the reviewer for this useful suggestion. While we did not perform negative-stain EM to directly visualize the liposomes, we did assess the quality and uniformity of the liposome preparation using dynamic light scattering (DLS). The DLS measurements confirmed an almost consistent diameter of 60nm, indicating successful liposome formation (Response Fig. 4).

Response Fig. 4. DLS profile of liposomes made from 100 % DOPG
(Acknowledgement: Suman Pal, IISER Pune, for his assistance)

- Line 478: Why was the concentration of OdinFtsZ1 and FtsZ2 different? Here the authors used more for FtsZ1 than FtsZ2, whereas in the 90° light scattering assay

they used less for FtsZ1 than FtsZ2. This is very counter intuitive and might introduce some concentration dependent patterns.

Response: Thank you for this observation. We agree that consistent concentrations are important for comparison. In response, we have updated **Fig. 1C** and **1D** to include light scattering profiles for OdinFtsZ1 and FtsZ2 at the same concentration to allow direct comparison [Page: 6-7, line no: 151-164].

The concentrations used in electron microscopy experiments were such that a good distribution of filaments were observed. We have not quantified or interpreted the images with respect to the number of filaments and their correlation with concentration. Rather, the difference in filament morphology for the two FtsZs has been the focus.

- Line 571: It would be good to have a list with the sequences of the used primers as well.

Response: Thanks, we have now included all the oligos sequences in the Structured methods- Reagents and tools table.

- All figures with gels: Please put in the figure legend what is the expected size of the different purified proteins. It is nowhere mentioned and should be mentioned at least once when you describe the proteins for the first time.

Response: Thanks for pointing this out, we have now included these details in the purity gel panels (**Appendix Fig. S2A** and **Fig. S3C**).

Referee #4:

In this work, Kumari, Uthaman et al. identify two FtsZ isoforms co-occurring together with tubulin in Odinararchaeota. The authors characterize differences in the polymerization and membrane tethering properties of the two FtsZ isoforms and investigate their divergent protofilament structures by cryoEM. While I am no expert in the field of eukaryotic cellular evolution, the comparison of the two FtsZ isoforms seems to provide insights into the evolution and diversification of the eukaryotic cytoskeleton.

I am serving as a technical reviewer for the cryoEM aspects of the manuscript. Generally, cryoEM imaging follows state-of-the-art procedures and cryoEM data seem of high quality as far as it can be judged from the data presented. However, analysis of the cryoEM data is very superficial, should be better described and most likely falls far behind what could be achieved with the available data.

There are several points that should be addressed in a revised version of the manuscript:

- 1) The authors should provide much more detail on cryoEM image analysis in the methods section. Also, it is good practice to provide a processing scheme in which the number of micrographs, the number of particles, the individual processing steps, software used etc. is being summarized. Please provide such a scheme as a supplementary figure.

Response: We thank the reviewer for this suggestion. Our initial analysis was limited to 2D classification and hence no processing workflow was provided. In the revised manuscript, we now include a 3D reconstruction of FtsZ1 and we have provided the cryoEM processing details for both OdinFtsZ1 and OdinFtsZ2 proteins in the method section [Page: 22-24]. Additionally, we have provided the workflow details for cryoEM processing for Odin FtsZ1 in **Fig. EV2**.

- 2) The authors should try to process their cryoEM data beyond the level of 2D classification and move on to 3D reconstruction, at least for FtsZ1. Their 2D classes indicate that at least a medium resolution 3D reconstruction could be achievable, which would be highly informative and substantially strengthen the manuscript.

Response: We sincerely thank the reviewer for this impactful suggestion, which has substantially strengthened the study. We now include 3D reconstruction of FtsZ1 combining 2D class averages from two different datasets at 0° tilt and 20° tilt which yielded a 3D reconstruction of the OdinFtsZ1 filaments at resolution of ~3.6Å (FSC=0.143) (**Fig. 2C-2E; Fig. EV3**). The details of sample preparation [Page: 22-24], data collection (**main Table: 1**) and results [Page: 7-9; line no: 166-223] are included in the revised manuscript.

- 3) Line 178: "Unlike the 2D class averages of OdinFtsZ1, which show high-resolution features, the 2D classes of OdinFtsZ2 show low-resolution features of the ring and no clear indication of subunit stoichiometry indicating heterogeneity." How do the authors come to this conclusion? At least based on what is shown in Fig. 2, the 2D classification for FtsZ1 and FtsZ2 has been performed at strongly differing pixel sizes and box sizes. FtsZ1 boxes contain only a few asymmetric units, while FtsZ2 particles are 'zoomed out' much more and contain entire rings of dozens of particles. One would likely not expect 2D classes of the quality as for

FtsZ1 at this level. The authors should repeat 2D classification for FtsZ2 using pixel sizes and box sizes as for FtsZ1.

Response: OdinFtsZ1 and FtsZ2 datasets were both collected at a pixel size of 1.07 Å. The box size for particle extraction was determined based on the particle dimensions. For FtsZ1, which forms filaments, a box size of 256 pixels was used to include several asymmetric units. In contrast, OdinFtsZ2 predominantly forms double rings measuring approximately 360–400 Å in diameter, for which a larger box size of 540 pixels was selected. The objective for OdinFtsZ2 was to obtain a 3D reconstruction of the complete double-ring structure, as this would offer valuable insights into its overall architecture.

Several other proteins also assemble into ring-like structures—such as DASH/DCM1 complex (Jenni et al 2018), nucleosomes (Bilokapic et al., 2018), dynamin-related protein 1 (DRP1) (Basu et al., 2017), and the Gasdermin pore (Johnson et al., 2024)—and in each case, 3D reconstruction has been successfully performed by averaging the full ring structure. This is feasible primarily because of the high degree of symmetry and homogeneity in these assemblies, where each ring is composed of a defined number of subunits in a consistent arrangement which enables reliable alignment and averaging of particles to achieve high-resolution reconstruction.

In contrast, various symmetries (C2, C24–C27) were applied for the FtsZ2 double-ring structures, but none yielded high-resolution reconstructions. This is likely due to heterogeneity in subunit number, both between adjacent rings and between the top and bottom rings within a single double-ring particle and will require sample optimisation and better images to obtain higher resolution reconstruction.

References :

1. Jenni, S., Harrison, SC., Structure of the DASH/Dam1 complex shows its role at the yeast kinetochore-microtubule interface. *Science*, 2018 360:552-558.
2. Bilokapic S, Strauss M, Halic M. Histone octamer rearranges to adapt to DNA unwrapping. *Nat Struct Mol Biol*. 2018 Jan;25(1):101-108.
3. Johnson, A.G., Mayer, M.L., Schaefer, S.L. et al. Structure and assembly of a bacterial gasdermin pore. *Nature* 628, 657–663 (2024).
4. Basu K, Lajoie D, Aumentado-Armstrong T, Chen J, Koning RI, Bossy B, Bostina M, Sik A, Bossy-Wetzels E, Rouiller I. Molecular mechanism of DRP1 assembly studied in vitro by cryo-electron microscopy. *PLoS One*. 2017 Jun 20;12(6):e0179397

4) Lines 163-165: Could the authors quantify the length of FtsZ1 filaments? In any case, it should become clear from the phrasing that the filaments in absence of any nucleotide are shorter as compared to the other conditions, which is not the case right now.

Response: We thank the reviewer for this suggestion. The average length of the OdinFtsZ1 filament is around 100 nm (average of 10 filaments) whereas in the absence of nucleotide, it is around 38 nm (average of 10 filaments).

5) Line 115: "Though OdinFtsZ2 clusters along with other FtsZ2 members (Fig. S1A), it possesses an insertion of a loop in the CTD, which is not found in other FtsZ proteins." Please support this claim by a multiple sequence alignment.

Response: We thank the reviewer for this suggestion. To support our claim regarding the unique loop insertion in the C-terminal domain (CTD) of OdinFtsZ2, we have now included a multiple sequence alignment of FtsZ2 proteins from various organisms, highlighting the presence of this loop insertion in OdinFtsZ2. Interestingly, we found that this loop insertion is also present in CetZ proteins, but it is absent in other FtsZ2 homologs. We have included the relevant alignment and have revised the text in the manuscript to refer to this new data (**Appendix Fig. S1A**) [Page: 6; line no: 147-150; Page: 15; line no: 414-416].

6) Lines 189-192: The authors point out a clear discrepancy between the pelleting behavior of FtsZ2 and its oligomeric state observed by EM. The authors should try to resolve this discrepancy or discuss in the text what could be possible reasons.

Response: We thank the reviewer for pointing this out. In characterization of polymerization proteins, assays such as pelleting and light scattering are more reliable when complemented by a direct observation using techniques such as negative stain EM since the species which are observed in the pellet need not always be filaments. We have observed cases when the filaments are short, they do not often pellet at 100,000 xg, but can be observed as a good distribution in an EM grid. This observation has now been described in the text. The differential sedimentation features indeed has been useful for us to study the interaction between FtsZ1 and FtsZ2 [page 9-10, lines 247-253]

- 7) The same centrifugal force and pelleting times are used in the filament pelleting and liposome pelleting assays. How can the authors differentiate pelleting due to liposome association from filament assembly in the liposome pelleting assay? In particular FtsZ1 was shown in this study to oligomerize independent from nucleotides. Please explain/discuss in the text.

Response: We thank the reviewer for this important comment. To clarify this, In the filament pelleting assay, the protein is directly subjected to ultracentrifugation, and the supernatant and pellet fractions are analyzed to assess filament formation.

In contrast, for the liposome pelleting assay, we take additional steps to differentiate liposome association from filament assembly. Specifically, we first subject the protein sample to ultracentrifugation to remove any pre-existing oligomers or filaments. The cleared supernatant—containing only monomeric or low-order oligomeric protein—is then incubated with liposomes, followed by a second ultracentrifugation step. This ensures that any protein detected in the pellet fraction results from liposome binding rather than sedimentation of pre-

formed filaments. This is also evident from the absence of protein in the pellet fraction in the absence of liposome (refer **Fig. 4B**, control lanes S and P as compared to the presence of the OdinFtsZ1 in the pellet fraction in **Fig. EV4A** (-GTP condition). We have now clarified this important distinction in the Methods and Results sections of the revised manuscript [page 20, lines 531-537]

Minor points:

1) Figs 2A-D need scale bars.

Response: Scale bar information for the micrographs are added on the images (**Fig. 2A, 2F**). For 2D class averages, the box and the pixel size information are provided in the figure legend.

2) For Fig S1, figure callouts in the text, panel labelling and figure captions/legends seem to be mixed up. Please correct.

Response: Corrected in the revised manuscript

3) Fig. 1A: please label the different FtsZ segments in the panel, instead of just color-coding them.

Response: Changed in the revised **Fig. 1A**

4) Fig. 2A,C: can the micrographs in Fig. 2 be shown at a similar contrast?

Response: As suggested by the reviewer, the contrast of the micrographs (Figure 2A, C) is adjusted to similar contrast using ImageJ.

5) Fig. 3: please label the identity of bands in SDS-PAGE gels.

Response: The expected molecular weight for each protein has been included in the SDS PAGE gels, along with the protein labels along the lanes (**Fig. S2A and S3C**).

6) Fig. 4A: red box not required?

Response: Thanks, this has been modified.

7) Lines 49-51: It is not quite clear what the authors refer to here. Do they refer to the growth direction as compared to tubulin? Please rephrase.

Response: Thanks, this has been modified, to provide a definition of the growth direction [page 3-4, lines 63-67]

8) Line 175: "In a given field of view, the number of longer stacked rings was less, but shorter stacks were more prominent." It is unclear to me what the authors relate to here. Do the authors refer to the number of rings per stacked assembly, the diameter of rings, etc? Please rephrase.

Response: This has been modified in the revised manuscript [page 9, lines 234-236].

Dr. Saravanan Palani
Indian Institute of Science
Biochemistry
Bangalore, Karnataka 560012
India

14th Jul 2025

Re: EMBOJ-2025-120418R
Distinct filament morphology and membrane tethering features of the dual FtsZs in Odinararchaeota

Dear Saravanan,

Thank you for submitting your revised manuscript to The EMBO Journal. Two of the original referees have now assessed it once more, and I am happy to say that they were generally satisfied with the revisions. Referee 2 still retains a few concerns regarding presentation, references, and evolutionary contextualization, which I would ask you to incorporate during a final round of minor revision. In addition, please also take care of the following remaining editorial issues:

- Please carefully go through the reference list and make sure that each reference is complete with citation year, volume, and page/locator numbers. DOIs (in conjunction with proper journal titles) should only be used in cases of pre-publications that do not have proper reference information yet.

- Please also adjust the format for citation of preprints as specified in our author guidelines:

The citation in the text should be: "(preprint: NAME1 et al, YEAR)"

The citation in the reference list: "Author NAME1, Author NAME2, ... (YEAR) article title. PREPRINT PLATFORM, doi: XXX"

- Please move the funding information included in the text into the Acknowledgement section, it should not be a separate subsection.

- In the Data Availability section, please add text URLs for directly accessing the databases (PDB, EMDB) in which datasets from the study have been deposited; and ensure that they are becoming publicly available at this point.

- Finally, I would suggest a minor change to the manuscript title, to avoid the somewhat awkward phrase "FtsZs":
Distinct filament morphology and membrane tethering features of the dual FtsZ paralogs in Odinararchaeota

I am returning the manuscript to you for a final round of minor revision, solely to allow you to make these modifications and upload the revised files. Once we will have received them, we should be ready to swiftly proceed with formal acceptance and production of the manuscript.

With kind regards,

Hartmut

- size of the scale bars that are mandatory for all micrograph panels
- the statistical test used to generate error bars and P-values

- the type error bars (e.g., S.E.M., S.D.)
- the number (n) and nature (biological or technical replicate) of independent experiments underlying each data point
- Figures may not include error bars for experiments with $n < 3$; scatter plots showing individual data points should be used instead.

9) To facilitate reproducibility and cross-laboratory adoption of methodologies, please structure the Materials & Methods section as outlined in our guide to authors, including a completed Reagents and Tools Table that can be downloaded from our author guidelines as well (<https://www.embopress.org/page/journal/14602075/authorguide#structuredmethods>).

10) Digital image enhancement is acceptable practice, as long as it accurately represents the original data and conforms to community standards. If a figure has been subjected to significant electronic manipulation, this must be clearly noted in the figure legend and/or the 'Materials and Methods' section. The editors reserve the right to request original versions of figures and the original images that were used to assemble the figure. Finally, we generally encourage uploading of numerical as well as gel/blot image source data; for details see: embopress.org/page/journal/14602075/authorguide#sourcedata

In the interest of ensuring the conceptual advance provided by the work, we recommend submitting a revision within 3 months (12th Oct 2025). Please discuss the revision progress ahead of this time with the editor if you require more time to complete the revisions. Use the link below to submit your revision:

Link Not Available

Referee #2:

This revised manuscript by Kumari et al. includes an improved presentation of the role of FtsZ in Odinararchaea. The authors have incorporated a better contextualization of their results in comparison with previous findings on archaeal systems, and have corrected most of the problematic discussion on the evolution of the cytoskeleton in the context of eukaryogenesis. I would like to congratulate the authors and recommend this manuscript for publication.

I only have a few very minor comments for the authors to consider:

- It is sometimes a little difficult to understand how references support the preceding sentence. I suggest the authors take another look at the chosen articles and question whether they are the best primary literature supporting a statement. For

example, in L37-39, the authors include 8 articles, of which 3 are reviews, 2 are very superficial evolutionary analyses (Liu et al 2021 and Zhang et al 2025), but missed the two most thorough phylogenomic studies on this question: Williams et al 2020 (Nat Ecol Evol) and Eme et al 2023 (Nature). Similarly, in L. 85-86, the authors include Santana-Molina et al 2023, which does not involve any study of SepF acting as a membrane anchor. This last example, may simply require rephrasing.

- L49-50, "prokaryotic" and "eukaryotic" are confusing terms here, since indeed the presence of the eukaryotic proteins in Asgard archaea make them, by definition, also prokaryotic. I would consider rephrasing.

- L80. "superphylum" should be "superphyla" (plural).

- The use of "Asgard phylum" is sometimes confusing. For instance, in L. 98, "Asgard phylum" is used to refer to its members, while in L 107 it is used to refer to the taxon. I would suggest to keep the latter meaning, and rephrase the former (and others throughout the text) as "Asgard archaea". Other instances of "Asgard" as an adjective could also be smoothed out, like "Asgard proteins" in L440 could read better as "Asgard archaeal proteins".

- L277-279. "The consistent presence of SepF in archaeal systems possessing FtsZ implies a strong functional interdependence between these proteins". There are probably many other proteins that are consistently present in archaeal systems possessing FtsZ for which there is absolutely no functional link. I suggest rephrasing.

- L410-412. This sentence reads a little strange.

- L437-439. Hodarchaeales are not proposed to be FECA, but the closest relative of FECA, based on Eme et al (2023, Nature). This should be corrected.

- L446-454. Information on the phylogenetic reconstruction is still missing. The authors should clarify how they selected sequences for the analysis, then how the sequences were aligned (and maybe trimmed), and then how they performed a phylogenetic reconstruction, following the analysis chronology. Moreover, the reconstruction aspect is still missing information to establish model selection (best models are indicated, but not what models were tested).

- Phylogeny figures. Indicating support values as circle sizes makes visualization extremely difficult, especially in trees where support values derive from ultrafast bootstraps and whose branches are therefore trustworthy at a very narrow range (>90-95%). Please include support values as numbers instead. Additionally, two different colors are chosen for Hadarchaeota (Hadarchaeota and Hadesarchaeota are the same group). And it would also be good if the figure legends indicate the number of alignment sequences and sites, the meaning of the support values (1000 ultrafast bootstrap replicates), and how they were rooted (in the case of the SepF tree).

- I won't press the authors to choose a different tree for EV1, but I would like to respond to their statement that the tree that ran under the simpler model has higher support values, for the sake of scientific discussion. Trees obtained under simple models can sometimes display higher support values irrespectively of their accuracy, since a poorly modeled tree can systematically recover artefactual clades in its replicates precisely due to poor modeling. Here, although the tree built from a complex model seems less congruent with known relationships, these are unsupported (e.g. position of Korarchaea, is separated by branches with values of 65% or less from its correct FtsZ clade). On the other hand, the tree with the simple model has relatively high support even for branches indicating incorrect monophylies (e.g. cyanobacteria and organellar FtsZ outside of bacteria, OdinFtsZ as sister to the rest of Asgard archaeal sequences for both FtsZ paralogs, a clade including members from different Euryarchaea and DPANN lineages). While some incorrect monophylies are also recovered by the complex model, these are generally less supported, indicative of slightly better interpretation of phylogenetic signal. However, I do understand if the authors prefer not to show a tree that might appear to break the monophyly of one of the FtsZ copies.

Referee #4:

The authors have done a great job in addressing my comments.

The additional data and analyses substantially strengthened the manuscript, which is now suitable for publication in The EMBO Journal.

EMBOJ2025-120418R1

Kumari et al., Distinct filament morphology and membrane tethering features of the dual FtsZ paralogs in Odinarchaeota

Point to Point Response to Referee

Referee #2:

This revised manuscript by Kumari et al. includes an improved presentation of the role of FtsZ in Odinarchaea. The authors have incorporated a better contextualization of their results in comparison with previous findings on archaeal systems, and have corrected most of the problematic discussion on the evolution of the cytoskeleton in the context of eukaryogenesis. I would like to congratulate the authors and recommend this manuscript for publication.

I only have a few very minor comments for the authors to consider:
- **It is sometimes a little difficult to understand how references support the preceding sentence. I suggest the authors take another look at the chosen articles and question whether they are the best primary literature supporting a statement. For example, in L37-39, the authors include 8 articles, of which 3 are reviews, 2 are very superficial evolutionary analyses (Liu et al 2021 and Zhang et al 2025), but missed the two most thorough phylogenomic studies on this question: Williams et al 2020 (Nat Ecol Evol) and Eme et al 2023 (Nature). Similarly, in L. 85-86, the authors include Santana-Molina et al 2023, which does not involve any study of SepF acting as a membrane anchor. This last example, may simply require rephrasing.**

Response: We thank the referee for the valuable suggestions. While some review articles were included based on earlier referee feedback, we agree that Williams et al. (2020) and Eme et al. (2023) are important primary references. These have now been added at L37–39, and the citations have been adjusted accordingly. For L85–86, we have revised the sentence to ensure that all references accurately support the corresponding statements.

- **L49-50, "prokaryotic" and "eukaryotic" are confusing terms here, since indeed the presence of the eukaryotic proteins in Asgard archaea make them, by definition, also prokaryotic. I would consider rephrasing.**

Response: We appreciate the referee's comments and have incorporated the suggested changes into the text. Line no – 49-53

- **L80. "superphylum" should be "superphyla" (plural).**

Response: Thank you, this has been corrected. Line no – 80

- The use of "Asgard phylum" is sometimes confusing. For instance, in L. 98, "Asgard phylum" is used to refer to its members, while in L 107 it is used to refer to the taxon. I would suggest to keep the latter meaning, and rephrase the former (and others throughout the text) as "Asgard archaea". Other instances of "Asgard" as an adjective could also be smoothed out, like "Asgard proteins" in L440 could read better as "Asgard archaeal proteins".

Response: This has been corrected as per the suggestions.

- L277-279. "The consistent presence of SepF in archaeal systems possessing FtsZ implies a strong functional interdependence between these proteins". There are probably many other proteins that are consistently present in archaeal systems possessing FtsZ for which there is absolutely no functional link. I suggest rephrasing.

Response: Thank you for the suggestion. We have now revised the statement. We now clearly mention that the consistent presence of SepF and FtsZ suggests a possible functional interdependence. Line no – 270-275

- L410-412. This sentence reads a little strange.

Response: This sentence has been rephrased. Line no – 410-413

- L437-439. Hodarchaeales are not proposed to be FECA, but the closest relative of FECA, based on Eme et al (2023, Nature). This should be corrected.

Response: We would like to point out that the L437-439 did mention that Hodarchaeales are the closest relatives to FECA. However, we have now added Eme et al which has reported the Heimdall ancestry of eukaryotes and Imachi et al which has recently reported the isolation of two strains of Hodarchaeales. Line no – 429-433

- L446-454. Information on the phylogenetic reconstruction is still missing. The authors should clarify how they selected sequences for the analysis, then how the sequences were aligned (and maybe trimmed), and then how they performed a phylogenetic reconstruction, following the analysis chronology. Moreover, the reconstruction aspect is still missing information to establish model selection (best models are indicated, but not what models were tested).

Response: Thank you for the comment. We have now expanded the methods section. Line no – 438-456

- Phylogeny figures. Indicating support values as circle sizes makes visualization extremely difficult, especially in trees where support values derive from ultrafast bootstraps and whose branches are therefore trustworthy at a very narrow range (>90-95%). Please include support values as numbers instead. Additionally, two different colors are chosen for Hadarchaeota (Hadarchaeota and Hadesarchaeota are the same group). And it would also be good if the figure legends indicate the number of alignment sequences and sites, the meaning of the support values (1000 ultrafast bootstrap replicates), and how they were rooted (in the case of the SepF tree).

Response: Thank you for the comment. We have revised Figures EV1 and EV5 as suggested by the referee, and the relevant details have been incorporated into the corresponding figure legends. Line no – 1130-1133; 1175-1177

- I won't press the authors to choose a different tree for EV1, but I would like to respond to their statement that the tree that ran under the simpler model has higher support values, for the sake of scientific discussion. Trees obtained under simple models can sometimes display higher support values irrespectively of their accuracy, since a poorly modeled tree can systematically recover artefactual clades in its replicates precisely due to poor modeling. Here, although the tree built from a complex model seems less congruent with known relationships, these are unsupported (e.g. position of Korarchaea, is separated by branches with values of 65% or less from its correct FtsZ clade). On the other hand, the tree with the simple model has relatively high support even for branches indicating incorrect monophylies (e.g. cyanobacteria and organellar FtsZ outside of bacteria, OdinFtsZ as sister to the rest of Asgard archaeal sequences for both FtsZ paralogs, a clade including members from different Euryarchaea and DPANN lineages). While some incorrect monophylies are also recovered by the complex model, these are generally less supported, indicative of slightly better interpretation of phylogenetic signal. However, I do understand if the authors prefer not to show a tree that might appear to break the monophyly of one of the FtsZ copies.

Response: We thank the referee for their comment and agree with them that sometimes simpler models might generate trees with greater support values because of systematic errors. However, the tree reported in the manuscript reproduces existing phylogenetic relationships more accurately. Hence, as pointed out by the referee, we wanted to avoid using a tree that is not congruent with the known relationships.